# The *careg* element reveals a common regulation of regeneration in the zebrafish myocardium and fin

Catherine Pfefferli[1] & Anna Jaźwińska[1]

The existence of common mechanisms regulating organ regeneration is an intriguing concept. Here we report on a regulatory element that is transiently activated during heart and fin regeneration in zebrafish. This element contains a *ctgfa* upstream sequence, called *careg*, which is induced by TGFβ/Activin-β signalling in the peri-injury zone of the myocardium and the fin mesenchyme. In addition, this reporter demarcates a primordial cardiac layer and intraray osteoblasts. Using genetic fate mapping, we show the regenerative competence of *careg*-expressing cells. The analysis of the heart reveals that the primordial cardiac layer is incompletely restored after cryoinjury, whereas trabecular and cortical cardiomyocytes contribute to myocardial regrowth. In regenerating fins, the activated mesenchyme of the stump gives rise to the blastema. Our findings provide evidence of a common regenerative programme in cardiomyocytes and mesenchyme that opens the possibility to further explore conserved mechanisms of the cellular plasticity in diverse vertebrate organs.

[1] Department of Biology, University of Fribourg, Chemin du Musée 10, 1700 Fribourg, Switzerland. Correspondence and requests for materials should be addressed to A.J. (email: anna.jazwinska@unifr.ch).

Adult zebrafish, axolotls and newts are valuable models in regenerative biology. These animals can survive substantial heart and limb injuries, and efficiently replace the lost parts with new functional tissues[1–10]. During reconstruction of the missing part, remnant injury-associated cells reactivate morphogenetic programs, proliferate and recreate a replicate of the original structure. Clonal analysis and genetic lineage tracing in salamanders and zebrafish have provided evidence that specialized cell types, such as muscle cells, cardiomyocytes and osteoblasts, undergo reversion to a less mature state, referred to as dedifferentiation, within the same lineage hierarchy[11–18]. Less-complex cells, such as epithelial, mesothelial and fibroblast-like cells, can more rapidly enter the regenerative programme, and additionally, may exhibit the plasticity to undergo transition to other developmentally related tissues or even more distant lineages through transdifferentiation[19]. The key regeneration-participating cells of the heart and the appendage, namely cardiomyocytes (CMs) and mesenchymal cells (MES), substantially differ in their architectural complexity, size and metabolism. An intriguing question is whether such diverse cell types activate common genetic regulatory mechanisms during the regenerative response.

Transgenic approaches in the zebrafish heart and fin showed that the regenerated tissues derive from local functional cells that can undergo limited dedifferentiation[12–15,17,18,20]. In the resected ventricle, analysis of the *gata4* transgenic reporter revealed that cortical CMs in the vicinity of the wound reactivate developmental gene expression and contribute to the new myocardium[13,21]. In the amputated fin, blastema formation requires the activation of MES in the stump. The distal part of the regenerative outgrowth acts as an apical signalling centre, whereas the proximal blastema comprises the proliferation and re-differentiation zone[22–26]. Chromatin-remodelling factors are required for both organ regeneration[27,28]. The lineage tracing and genetic clonal analyses have provided no evidence for a contribution of remote stem cells to the heart and fin regenerate[12,13,20,29].

Despite the anatomical differences between the heart and fin, a concept of common regulatory mechanisms of regeneration has been recently proposed. This idea is based on the identification of a short DNA sequence upstream to the zebrafish metabolic gene, *leptin b*, which is induced in the endocardium and wound tissue of the heart, as well as in the blastema of the fin[30]. No activation of this enhancer element, however, has been observed in CMs. Thus, it still remains unknown whether common regulatory elements exist for injury-responsive cardiac myocytes and fin fibroblasts.

The ventricle of the adult zebrafish heart is a complex structure, containing an outer cortical (compact) layer and inner trabecular (spongy) fascicles[31]. Both types of myocardium are separated by flattened cardiac cells, called junctional (transitional) CMs[32,33]. On the basis of multicolour clonal analysis, this cardiac layer has been named 'primordial' layer, which is established in the early embryonic heart and maintained throughout ontogenesis[34]. While the *gata4*-reporter studies have provided evidence for a regenerative contribution of the cortical myocardium[13], the restorative potential of trabecular and primordial CMs remains unclear.

In this study, we identified a 3.18 kb DNA fragment containing an upstream genomic sequence of the *ctgfa* gene, called the *careg* element, which is activated in the regenerating ventricular CMs and fin MES in a TGFβ/Activin-β signalling-dependent manner. Our findings yield insight into the mechanisms of heart and fin restoration and provide evidence for the existence of regeneration regulatory programs that are conserved between architecturally diverse cells, such as cardiomyocytes and mesenchymal cells.

## Results

**A reporter of peri-injured cardiomyocytes and fin mesenchyme.** To uncover the mechanisms underlying the injury-dependent reactivation of morphogenetic programs in distinct organs, we searched for an informative DNA regulatory element that is induced in regenerating myocardium and fin. In both organs, injury-activated tissues upregulate synthesis of specific developmental proteins. In the heart, a portion of the trabecular myocardium within 100 μm from the injury border reactivates the expression of the embryo-specific cardiac myosin heavy chain isoform, embCMHC[35]. In the fin, MES reorganization occurs within at least 300 μm from the amputation plane, which is associated with the upregulation of Tenascin C, a de-adhesive extracellular matrix protein[36]. Using a candidate approach, we identified a transgenic reporter that overlaps with the transient expression of embCMHC and Tenascin C in the regenerating hearts and fins, respectively. The reporter was driven by a 3.18 kb upstream regulatory sequence of *connective tissue growth factor a* (*ctgfa*)[37]. This transgenic line was originally established to investigate notochord development, and was also used to highlight the connective tissue of regenerating fins[37,38], but it had not yet been investigated in the heart. Thus, the activation of *ctgfa:EGFP* in the embCMHC-positive region was an unexpected finding.

To determine the spatiotemporal pattern of the reporter activity during organ regeneration, we analysed the dynamics of *ctgfa:EGFP* after ventricular cryoinjury and fin amputation. In uninjured hearts, this reporter was detected in a fine sheet of cells along the ventricular circumference (Supplementary Fig. 1a,b). At 1 d.p.ci. (days post-cryoinjury), the injured myocardium was recognized by the absence of *ctgfa:EGFP* expression in the outer heart layer, as opposed to the intact part of the ventricular wall (Fig. 1a,b). No *ctgfa:EGFP* expression was observed in the trabecular myocardium (Fig. 1c). At 4, 7 and 14 d.p.ci., fluorescence persisted in the subcortical layer, and additionally, it emerged in ~60% of the cardiac muscle within a distance of 100 μm from the injury border (Fig. 1d–f; Supplementary Fig. 1c–f). Approximately 20% of these cells also expressed embCMHC at 7 and 14 d.p.ci. (Fig. 1f; Supplementary Fig. 1i). Towards the end of the regenerative process at 30 d.p.ci., the expression of the reporter became reduced, and it was nearly undetectable within the new myocardium at 60 d.p.ci. (Fig. 1g,i; Supplementary Fig. 1g,h). We concluded that *ctgfa:EGFP* represents a reliable marker of adult CMs that transiently reactivate the regenerative programme in the peri-injured myocardium.

Consistent with previous studies of zebrafish fin regeneration[38], live-imaging of caudal fins revealed that *ctgfa:EGFP* was strongly upregulated in the regeneration zone at 1 and 3 d.p.a. (days post amputation), and declined when the restorative process was completed at 20 d.p.a. (Fig. 1j–o; Supplementary Fig. 2a–l). At 1 and 3 d.p.a., the regeneration zone included not only the outgrowing blastema, but also the peri-injury stump tissue within a distance of three ray segments below the amputation plane. Immunofluorescence analysis of longitudinal fin sections at 3 d.p.a. showed that the expression of *ctgfa:EGFP* was induced in the Tenascin-C-positive MES of the stump and the blastema (Fig. 1q). No expression was observed in dedifferentiated osteoblasts above the injury plane (Fig. 1s). Little fluorescence was detected in the distal blastema (Supplementary Fig. 2m,n). Taken together, *ctgfa:EGFP* is transiently upregulated during dedifferentiation in both CMs and MES at the injury border of the heart and the fin, respectively.

Although *ctgfa:EGFP* was strongly induced in the regeneration zone, we observed its homeostatic expression in the intact heart

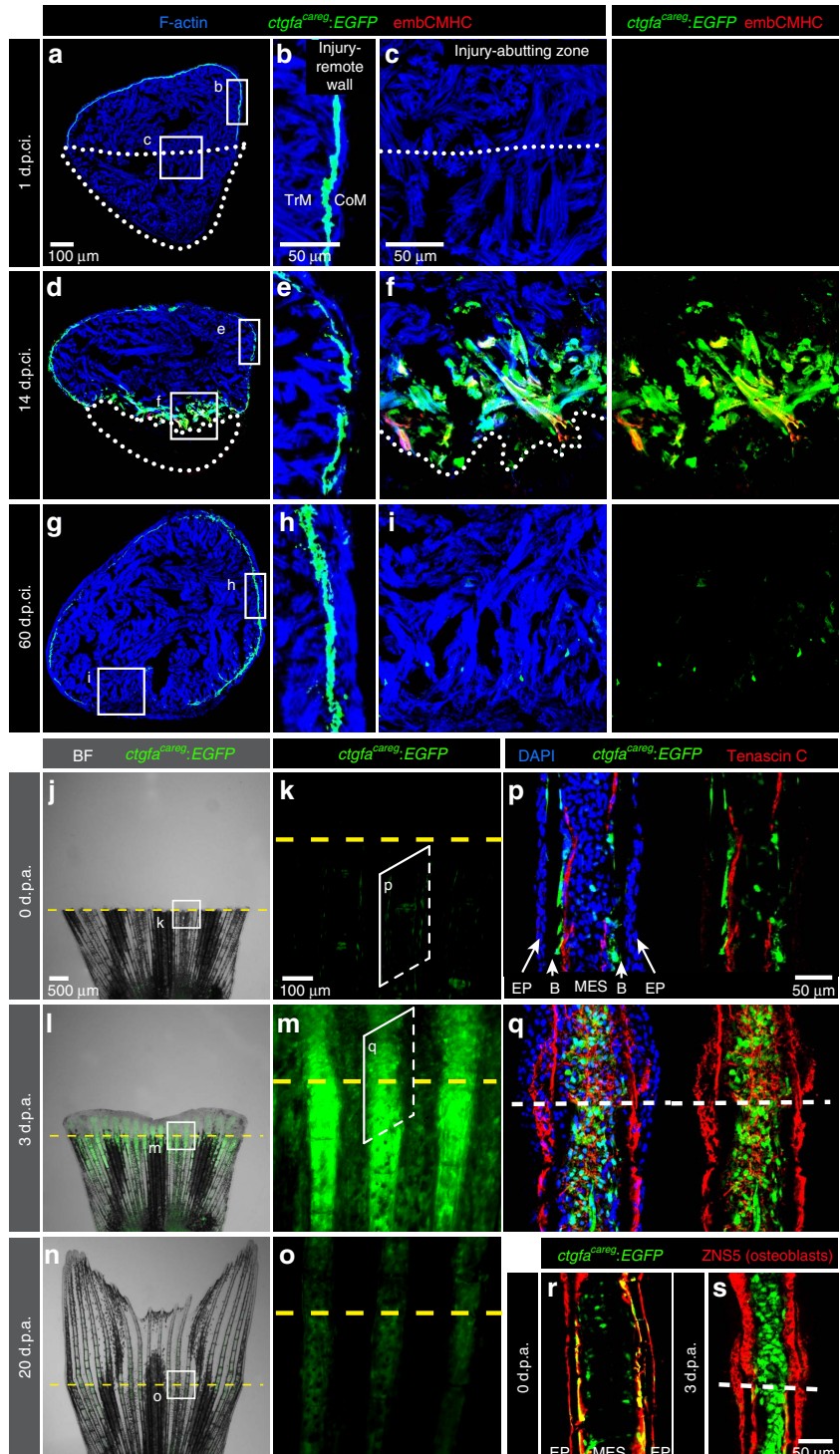

**Figure 1 | A transgenic reporter of the peri-injured myocardium and the fin stump.** (**a**–**i**) Transversal ventricle sections of transgenic fish *ctgfa^careg^:EGFP* immunostained for embCMHC (red) at different days post-cryoinjury (d.p.ci.). The cardiac muscle is detected by F-actin staining (Phalloidin, blue). (**b**,**e**,**h**) The injury-remote part of the ventricular wall displays a subcortical layer of *ctgfa^careg^:EGFP* cells that is located between the thin cortical myocardium (CoM) and the inner trabecular myocardium (TrM). This layer does not express embCMHC and remains unaltered during regeneration. (**c**,**f**,**i**) The injury-abutting zone of the ventricular wall shows transient expression of *ctgfa^careg^:EGFP* and embCMHC within a distance of 100 μm from the wound border during regeneration. *N* ≥ 4. (**j**–**o**) Live-imaging of fins at different days post amputation (d.p.a.). Bright-field (BF) was combined with fluorescence. (**k**,**m**,**o**) Higher magnifications of the region at the amputation plane (yellow dashed line) show transient expression of *ctgfa^careg^:EGFP* in the stump and the regenerate. *N* = 4. (**p**–**s**) Immunofluorescence staining of longitudinal fin sections. (**p**,**r**) At 0 d.p.a., *ctgfa^careg^:EGFP⁺* cells are associated with Tenascin C fibres along the border between the bones (B) and mesenchymal cells (MES), but not the epidermis (EP). Some of these cells express an osteoblast marker visualized with Zns5 antibody (red). (**q**,**s**) At 3 d.p.a., Tenascin C demarcates the zone of tissue remodelling in the stump and the outgrowth. *ctgfa^careg^:EGFP* is upregulated in the activated MES at the peri-injury zone, but not in Zns5-labelled osteoblasts of the outgrowth. *N* ≥ 4. Post-infarcted ventricle is encircled with a dotted line. Fin amputation plane is shown with a dashed line. The same rules apply to all subsequent figures.

and fin, namely in the subcortical layer along the ventricular circumference and in cells along the interface between the bones and MES, respectively (Fig. 1b,e,h,p). In intact fins, *ctgfa:EGFP*-expressing cells colocalized with Zns5-labelled osteoblasts, predominantly at the interface between acellular bone matrix and MES in the intrarays (Fig. 1r). These cells were associated with extracellular de-adhesive Tenascin C, suggesting local tissue remodelling between MES and the bones (Fig. 1p,r). Thus, *ctgfa:EGFP* is not solely a marker of injury-abutting cells, but also of certain junctional cells located at tissue borders during homeostasis.

**A distinct expression of *ctgfa:EGFP* and endogenous *ctgfa*.** To determine whether the *ctgfa:EGFP* reporter matches the endogenous expression pattern of the *ctgfa* gene, we performed *in situ* hybridization on sections of hearts at 14 d.p.ci. and fins at 3 d.p.a. Unexpectedly, we found that anti-sense probes against *ctgfa* and *enhanced green fluorescent protein* (*egfp*) labelled different cells (Fig. 2a–h). *ctgfa* mRNA was present solely in the wound tissue of the heart, and in cells associated with the bones of fins (Fig. 2a,b,e,f). By contrast, *egfp* was detected in CMs along the ventricular circumference and in the vicinity of the injury of the heart, and in the blastema of the fin regenerate (Fig. 2c,d,g,h). Thus, in both organs, the *egfp* mRNA distribution reproduced the pattern observed by fluorescence emission, while the endogenous *ctgfa* gene had a distinct expression pattern. We concluded that the *ctgfa:EGFP* transgene contains a sequence that is regulated in a unique manner in the peri-injury zone of regenerating organs. Accordingly, we renamed this reporter, '*careg*' (*ctgfa* reporter in *reg*eneration), to distinguish it from the endogenous locus that includes additional genomic sequences regulating *ctgfa* mRNA expression. Below, we used *careg* to refer to this reporter.

To verify that the specificity of the *careg:EGFP* expression pattern is related to the 3.18 kb *ctgfa* upstream sequence and not to the genomic integration site of the construct, we created another transgenic line. We used a fluorescent protein with a rapid turnover, namely destabilized monomeric Kusabira-Orange 2 (dmKO2), which contains a C-terminal PEST domain targeted for rapid degradation via ubiquitinylation[39]. Double transgenic fish, *careg:EGFP;careg:dmKO2* displayed an overlapping pattern in regenerating CMs and MES (Fig. 2i–m). The dynamics of *careg:dmKO2* during heart and fin regeneration was similar to that of *careg:EGFP* (Supplementary Figs 3,4). On the basis of the same expression pattern of both constructs, we concluded that *careg* indeed contains a regulatory element that is specifically activated during regeneration.

To determine the features of the *careg* sequence, we performed pairwise alignment of the *ctgfa* genomic region between zebrafish and 9 other fish species from distinct phylogenetic orders, as well as with *Xenopus*, chicken, mouse and human (Fig. 3a,b). This analysis identified two conserved regions within the *careg* sequence, namely, a 150 bp fragment immediately before the transcription start site and a 400 bp region at ~1 kb upstream from this position (Fig. 3b). The latter is conserved in five other fish species from different orders, namely cavefish, spotted gar, tilapia, platyfish and a lobe-finned fish Coelacanth (*Latimeria*). Interestingly, this region was also present in Xenopus and chicken, but not in mammals or in certain fish, such as medaka, fugu, stickleback or tetraodon. Bioinformatics analysis of the entire *careg* sequence predicted several different transcription factor-binding sites, such as Smad3, TCFF, MEF3, NKXH and GATA (Fig. 3c; Supplementary Data 1). We concluded that the *careg* element consists of a combination of evolutionary conserved and unique sequences that might be regulated by multiple transcription factors.

**The *careg* reporter in various injury models.** To test whether the *careg* reporter is activated in other injury models, we switched between the procedures used for each organ, namely, we performed amputations of the ventricular apex and cryoinjury of the caudal fin[40,41]. Examination of longitudinal heart sections at 7 d.p.a. revealed that *careg:EGFP* was induced within 100 μm in the peri-injury zone of the myocardium, which also upregulated embCMHC expression (Fig. 4a,b). Thus, both methods, cryoinjury and resection, induce a local activation of *careg:EGFP* in CMs along the damaged border.

As opposed to the fin amputation model, the non-surgical exposure to the cold blade along the cryoinjury plane results in a progressive and spontaneous tissue detachment that is apparent between 12 and 48 h after procedure[41]. Analysis of fins shortly after cryoinjury did not reveal any expression of *careg:EGFP* (Fig. 4c,d). During shedding of the damaged distal part, at 1 and 2 d.p.ci., *careg:EGFP* was weakly induced in the stump, suggesting that the clearance of dead tissues was associated with a concomitant activation of the regenerative programme in the remaining cells (Fig. 4e,h). Remarkably, the expression of *careg:EGFP* became further upregulated at 4 d.p.ci., when the regenerative process was resumed after reparation of the distorted margin (Fig. 4i,j). As in amputated fins, *in situ* hybridization on longitudinal fin sections showed that the expression of the reporter was not associated with the induction of the endogenous *ctgfa* gene within the activated MES (Supplementary Fig. 5a–d). At the advanced regenerative phase, at 14 d.p.ci., *careg:EGFP* was downregulated, consistent with the amputation model (Fig. 4k,l).

To further test the regenerative response of the *careg* element, we performed a regeneration assay of the larval tail, in which the posterior tip of the notochord was truncated. Consistent with previous *ctgfa* studies[37], *careg:dmKO2* displayed a basal expression level in the larval notochord (Supplementary Fig. 5e). The amputation site, however, transiently upregulated *careg:dmKO2* levels during regeneration (Supplementary Fig. 5f–i). Taken together, the *careg* reporter is activated in regeneration-responsive cells in several injury models, suggesting its general suitability for studying organ regeneration in zebrafish.

**The activation of *careg* is dependent on TGFβ and Activin-β.** Both heart and fin regeneration requires TGFβ/Activin-β signalling[36,42]. In the fibrotic tissue of the injured ventricle, this pathway also induces a transient deposition of certain matrix components, such as a remodelling protein Tenascin C, fibril-forming Collagen I and fibril-associated Collagen XII (refs 42,43). To investigate whether TGFβ/Activin-β signalling is active in the *careg* expression zone of regeneration-activated CMs and MES, we analysed phosphorylation of Smad3, which is a canonical downstream signal transducer of this pathway[44]. Ventricle sections at 7 d.p.ci. revealed that $57.3\% \pm 7.1$ ($N=6$) of *careg:EGFP*-positive cells contained pSmad3-labelled nuclei (Fig. 5a–c). Fin sections at 3 d.p.a. showed $70.3\% \pm 1.8$ ($N=7$) of *careg:EGFP*/pSmad3 double-positive cells in the MES compartment at 200 μm below the amputation plane (Fig. 5o,p). This suggests that the peri-injury zone of both organs is substantially activated by TGFβ/Activin-β signalling.

To test the requirement of this pathway for *careg* expression, we used the pharmacological inhibitor of the TGFβ/Activin-β type I receptor, SB431542 (refs 36,42). In uninjured heart, the prolonged treatment for 7 days with this drug did not affect the homeostatic expression of *careg:EGFP* in the outer layer of the ventricle (Supplementary Fig. 6a–c). Consistently, at 7 and 14 d.p.ci., the inhibition of the TGFβ/Activin-β pathway did not display any effect on reporter expression along the intact myocardial wall of *careg:EGFP* and *careg:dmKO2* hearts. By

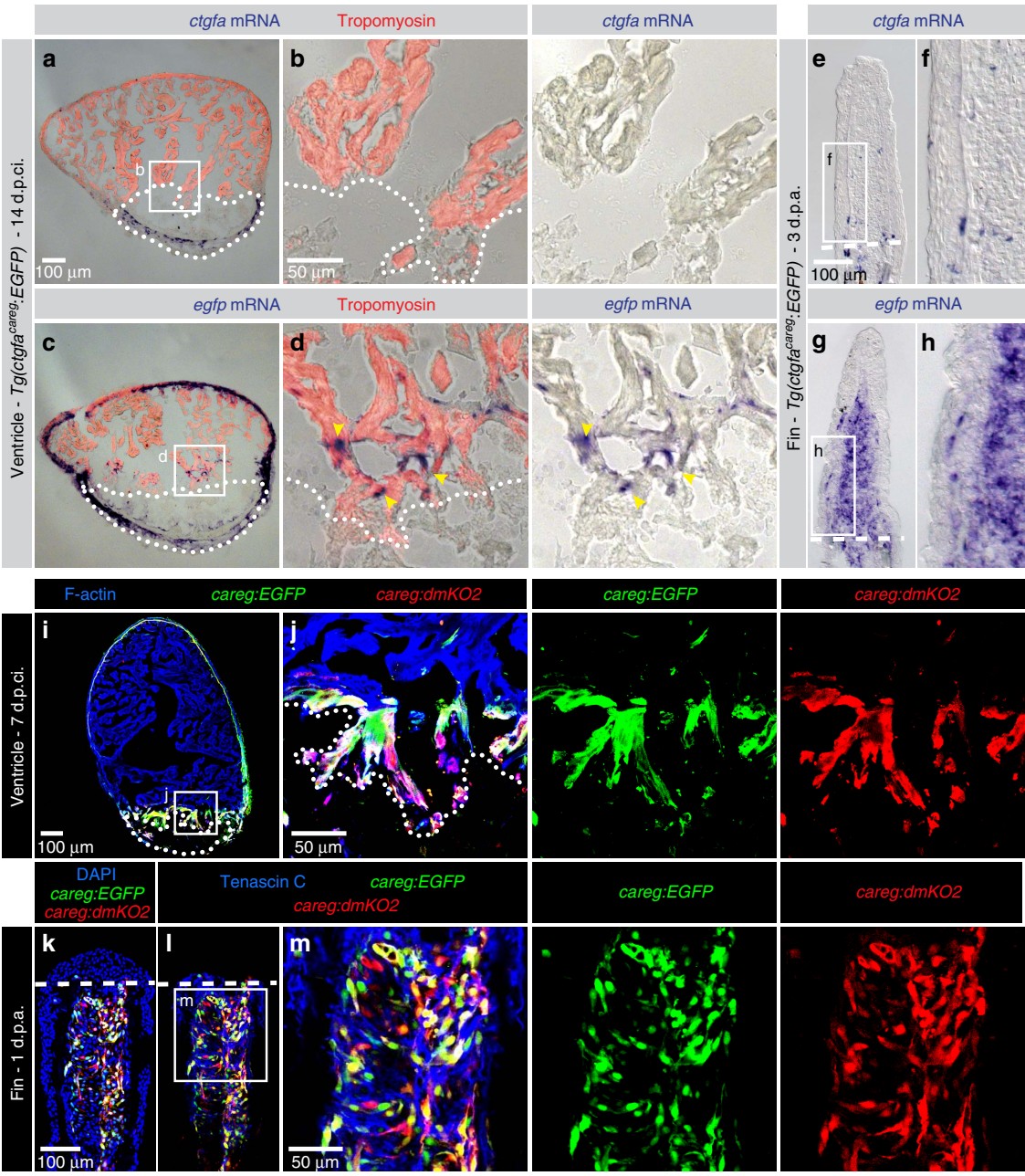

**Figure 2 | ctgfa^careg:EGFP does not reproduce endogenous ctgfa expression in peri-injury tissues.** (a–h) *In situ* hybridization using *ctgfa* and *enhanced green fluorescent protein (egfp)* probes on sections of hearts at 14 d.p.ci. and fins at 3 d.p.a. of *ctgfa^careg:EGFP* transgenic fish shows distinct expression patterns. (**a,b**) *ctgfa* mRNA is not detected in the myocardium, but in a subset of non-myocytes in the post-infarcted tissue. (**c,d**) *egfp* mRNA is expressed in the outer wall of the myocardium and at the peri-injury zone (yellow arrowheads). (**e,f**) *ctgfa* mRNA is present in a few bone-associated cells of the fin, while (**g,h**) *egfp* mRNA is detected in the blastema. N = 6. (**i–m**) *careg:EGFP* and *careg:dmKO2* double transgenic fish display an overlapping expression pattern in the regenerating heart at 7 d.p.ci. and the fin at 1 d.p.a. N = 4.

contrast, the peri-injury myocardium showed a three- to fourfold reduction of *careg*-labelled CMs (Fig. 5d–h, Supplementary Fig. 7a–j). This phenotype was also associated with attenuated embCMHC immunolabelling. We concluded that the regenerative induction of the *careg* element and the endogenous embCMHC expression in the peri-injured myocardium is regulated by the TGFβ/Activin-β signalling pathway.

Restorative cardiogenesis requires CM proliferation. To determine the effect of SB431542 treatment on CM proliferation specifically in the *careg*-positive regenerating myocardium, we assessed BrdU incorporation within the region of 100 μm from

the injury border. This analysis revealed a fourfold decrease in the number of Mef2c/BrdU double-positive cells after inhibitor treatment (Supplementary Fig. 7k–o). Thus, the inhibition of TGFβ/Activin-β signalling affects not only dedifferentiation, but also proliferation of CMs in the peri-injury zone.

Next, we analysed the regulation of *careg:EGFP* in regenerating fins. In uninjured fins, the prolonged treatment for 7 days with SB431542 did not affect the homeostatic expression of *careg:EGFP* in osteoblasts (Supplementary Fig. 6d,e). At 1 and 3 d.p.a., however, the inhibition of TGFβ/Activin-β strongly prevented the regenerative outgrowth formation[36], and the

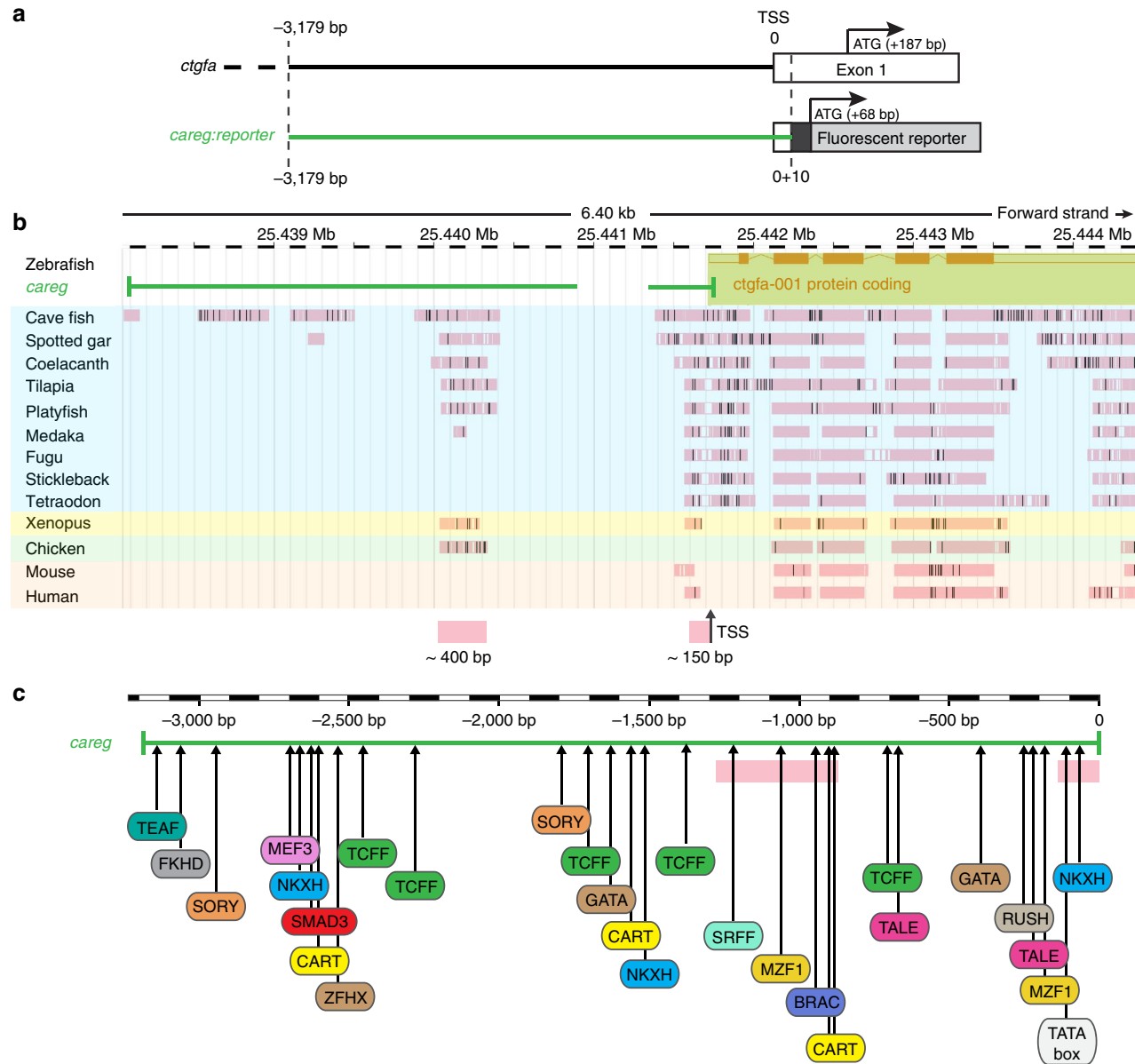

**Figure 3 | Sequence analysis of the *careg* element. (a)** Comparison of the *careg* element and the *ctgfa* genomic sequence of wild-type AB strain zebrafish, which was used to generate transgenic fish. The *careg* element includes a 3,179 bp sequence immediately upstream of the transcriptional start site (TSS) and 10 bp of the 5′-untranslated region of *ctgfa*. **(b)** LASTz net pairwise alignment of the *ctgfa* genomic region between Zebrafish (*Danio rerio*, Ensembl version 87.10, Chromosome 20; 25′438′500 - 25′444′700) and 9 other fish species (highlighted in blue), Xenopus (yellow), chicken (green) and 2 mammals (orange). The pink boxes indicate the conserved regions in the majority of the species. Black and white bars within the pink boxes represent gaps in the alignments. The *careg* sequence is indicated with a green line above the alignment. The gap in the *careg* sequence corresponds to additional 465 bp in the sequence from the Ensembl database. This sequence is absent in the genome of AB zebrafish strain. The scale bar indicates the genomic position in the chromosome 20 of zebrafish. Two conserved regions of ∼400 and 150 bp are indicated with pink boxes at the bottom of the alignment. **(c)** Prediction of transcription factor (TF) binding sites in the *careg* sequence (green) with MatInspector (Genomatix). The pink boxes indicate the conserved regions identified in **b**. A SMAD3 binding site is highlighted in red. The complete list of TF binding sites is given in Supplementary Data 1.

activation of *careg:EGFP* in the stump (Fig. 5i–l; Supplementary Fig. 8a–d). To determine whether *careg:EGFP* expression was suppressed due to a general regenerative failure, we tested the *fgf20a* mutant fish, called *dob*, which fail to form a blastema[45]. Despite an absence of the regenerative outgrowth, *dob* mutant fins normally induced *careg:EGFP* in the peri-injured stump at 1 and 3 d.p.a. (Fig. 5m,n; Supplementary Fig. 8e–f). We concluded that the regulation of the *careg* reporter in the fin is dependent on TGFβ/Activin-β signalling, irrespectively of the ability to form a blastema.

Although Fgf20a and TGFβ/Activin-β signalling have been described as the early inducers of blastema formation, the interaction between both pathways remains unknown. Using the *careg* element and pSmad3 immunofluorescence, we addressed the question whether the function of TGFβ/Activin-β signalling is dependent on Fgf20a. Our analysis of longitudinal fin sections revealed that the pSmad3 activity was blocked after SB431542 treatment, but remained unchanged in *dob* mutant fins (Fig. 5o–t). This finding indicates that TGFβ/Activin-β acts independently of the Fgf20a activity during fin regeneration, providing a

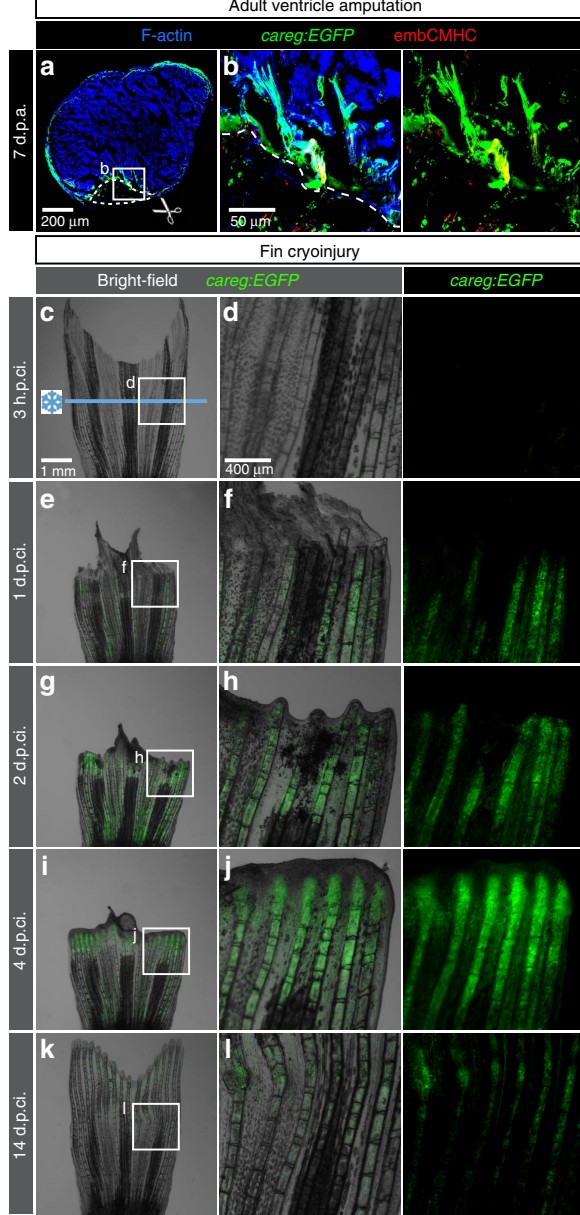

**Figure 4 | The *careg* reporter is activated in other injury models of the zebrafish heart and fin.** (**a**,**b**) Ventricle section of *careg:EGFP* heart at 7 days post amputation (d.p.a.) labelled with F-actin (blue) and antibodies against GFP (green) and embCMHC (red). *careg:EGFP* expression is activated in the ventricular trabeculae close to the amputation plane of the resected apex (dashed line). N = 3. (**c–l**) Live-imaging of the same fin at different time points post-cryoinjury. Cryoinjury of the caudal fin results in spontaneous sloughing of destroyed tissue within two days after the damage induction, followed by resumed regeneration. Bright-field was combined with fluorescence. (**c**,**d**) At 3 h.p.ci. (hours post-cryoinjury), *careg:EGFP* is not detected in the fin. Ischaemic tissue, which is distal to the cryoinjury plane (blue line), remains integrated with the rest of the body at this early stage. (**e**,**f**) At 1 d.p.ci., the majority of the damaged tissue detached from the stump. *careg:EGFP* is detected proximally to the damaged zone. (**g**,**h**) At 2 d.p.ci., the reparation of the distorted margin is accompanied by *careg:EGFP* expression. (**i**,**j**) At 4 d.p.ci., the protruding blastema displays enhanced expression of the *careg* reporter. (**k**,**l**) The advanced regeneration is associated with downregulation of *careg:EGFP* expression. N = 3.

new insight into the molecular network guiding blastema formation.

Taken together, the analysis of hearts and fins revealed that the same molecular pathway, TGFβ/Activin-β, stimulates the upregulation of *careg* in distinct regeneration-participating tissues, such as CM and MES, but not in the fibrotic tissue.

**careg in primordial CMs during heart development.** Heart regeneration has been proposed to involve a reactivation of cardiac embryonic programs[3–5]. To determine whether the induction of the *careg* reporter in the peri-injured CMs demarcates less differentiated CMs, we performed analysis of the embryonic heart. We found that at 1 and 3 days post-fertilization (d.p.f.), *careg:EGFP* and *careg:dmKO2* fully colocalized with embCMHC in embryonic CMs (Fig. 6a–f; Supplementary Fig. 9a,b). This finding suggests that the *careg* regulatory element represents a common molecular marker of activated CMs during regeneration and immature CMs during development.

To investigate the reporter expression in subsequent developmental stages, we crossed *careg:EGFP* with the *cmlc2:DsRed-nuc* line, which labels cardiac nuclei. In larvae at 12 d.p.f., we found that *careg* expression persisted in CMs of the heart wall, and a few *careg*-expressing cells were scattered in the ventricular chamber (Fig. 6g–i; Supplementary Fig. 9c). These inner CMs probably detached from the heart surface in the process of trabeculation, which involves delamination of CMs from the outer wall into the heart chamber[46]. Consistently, after completion of the trabeculation at 30 d.p.f., *careg:EGFP* and *careg:dmKO2* were restricted to the outer myocardial wall (Fig. 6j–o). This result indicates that the reporter expression labels the primordial cardiac wall throughout development.

To genetically test the hypothesis that *careg*-positive CMs in the larval heart contribute to the delamination process of trabecular CMs, we performed lineage tracing of these cells. We generated a transgenic line with a tamoxifen-dependent Cre recombinase driven by the *careg* regulatory sequence, named *Tg(careg:CreERT2)*. We crossed this strain with the *ubiquitin:-loxP-EGFP-STOP-loxP-mCherry* lineage-tracing line, referred to as *ubi:Switch,* which marks cells with red fluorescence after Cre-lox mediated recombination (Fig. 7a). To label embryonic *careg*-positive cells, we pulsed embryos at 75% epiboly stage with 4-hydroxytamoxifen (4-OHT) for two days and analysed the hearts at 21 d.p.f. (Fig. 7b). We found that nearly 60% of CMs (58.2 ± 2.5; N = 9) within both the trabecular and outer myocardium were randomly demarcated by mCherry expression, whereas no mCherry was observed in embryos treated with the vehicle (Fig. 7c). This fate-mapping result reveals that *careg*-positive embryonic CMs contribute to the trabecular myocardium during development.

**careg in the primordial/junctional layer of adult hearts.** Next, we examined both fluorescent transgenic reporters, *careg:dmKO2* and *careg:EGFP,* in the outer wall of the adult heart. Heart sections revealed a strong co-expression of the fluorescent proteins in a single subcortical cell layer between the compact and trabecular myocardium (Fig. 8a–c). This cell layer co-expressed *cmlc2:EGFP* and Tropomyosin, confirming the cardiac identity. Considering developmental expression of *careg* and the recent multicolour clonal analysis[34], we concluded that the *careg* reporter is a marker of the 'primordial' CMs in the zebrafish ventricle.

To determine the role of the primordial layer in the adult heart, we analysed features of the *careg*-expressing CMs. First, Tropomyosin revealed a reduced pattern of sarcomeres in

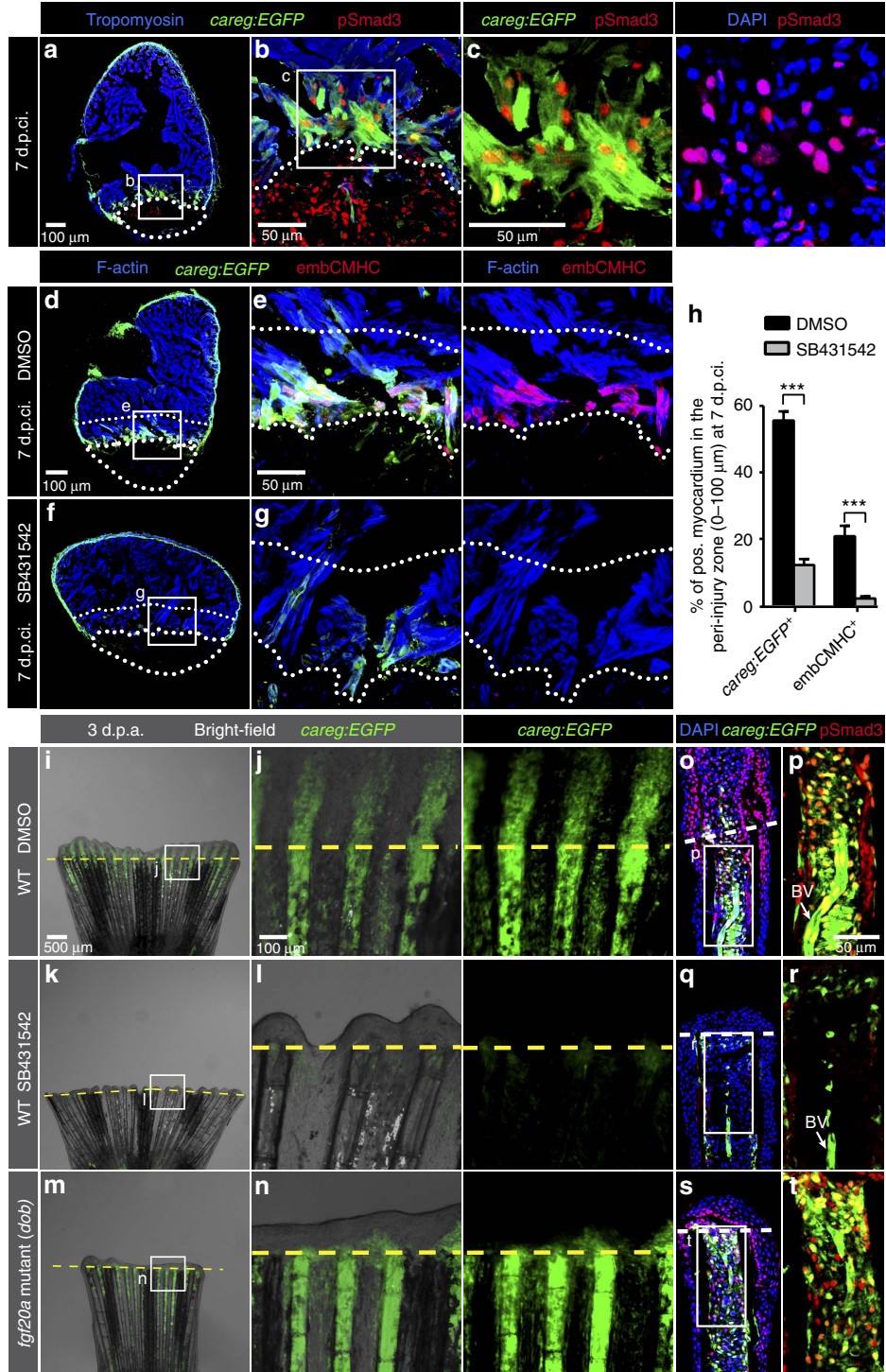

**Figure 5 | The regeneration biosensor *careg* is dependent on TGFβ/Activin-β signalling.** (**a–c**) Immunofluorescence staining of *careg:EGFP* ventricle at 7 d.p.ci. with antibodies against GFP (green), pSmad3 (red) and Tropomyosin (blue) revealed the presence of TGFβ/Activin-β-activated cells in the *careg:EGFP*-expressing tissue. N = 6. (**d–g**) *careg:EGFP* heart sections at 7 d.p.ci. treated with 0.1% DMSO or 20 μM SB431542, an inhibitor of TGFβ type I receptors, and immunostained with antibodies against GFP (green) and embCMHC (red). The intact myocardium was detected with F-actin staining (blue). In the magnified images, the upper dotted line demarcates a 100 μm-thick margin of the remaining myocardium from the injury border (lower dotted line). (**h**) Percentage of *careg:EGFP*⁺ and embCMHC⁺ area within a distance of 100 μm from the post-infarcted tissue in hearts at 7 d.p.ci. treated with DMSO or SB431542. The inhibitor treatment resulted in a significant reduction of *careg:EGFP* and embCMHC expression in the peri-injured zone, compared to control hearts treated with DMSO. N = 8. ***P < 0.001; unpaired t-test. Error bars correspond to s.e. of the mean (s.e.m.). (**i–n**) Live-imaging of *careg:EGFP* fins at 3 d.p.a. in a wild-type (WT) background treated with DMSO or SB431542, and in the *fgf20a* (*dob*) mutant background. *careg:EGFP* expression is suppressed in the stump by TGFβ inhibition, but it remains normal in *fgf20a* mutant fins. (**o–t**) Longitudinal sections of the fins shown in the left panels display a reduction of pSmad3-positive nuclei (red) and *careg:EGFP* in the stump of SB431542-treated fins, whereas both markers are unaltered in *fgf20a* mutants. BV, blood vessel. N = 4.

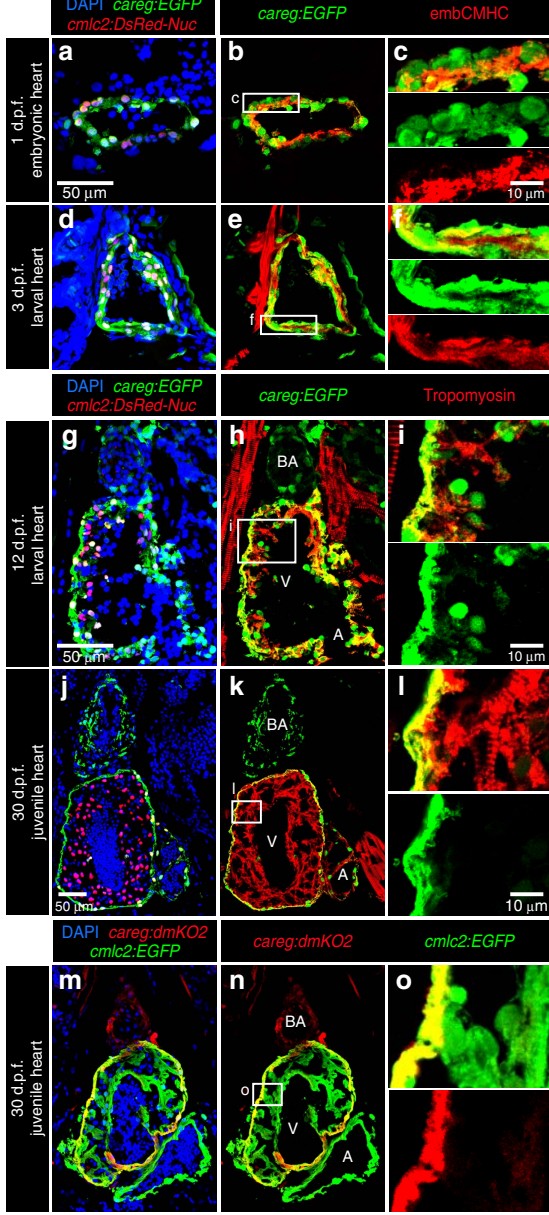

**Figure 6 | *careg* is expressed in embryonic CMs and the outer wall of developing ventricle.** (**a–l**) Longitudinal sections of *careg:EGFP;cmlc2:DsRed2-Nuc* double transgenic hearts at different time points during development. Cardiac nuclei are marked by DsRed expression. The endogenous fluorescence was quenched with HCl treatment before immunostaining. GFP and DsRed were detected by antibody staining. (**a–f**) At 1 and 3 d.p.f. (days post-fertilization), *careg:EGFP* and endogenous embCMHC are co-expressed in embryonic CMs. (**g–i**) At 12 d.p.f., a few *careg:EGFP⁺* CMs delaminate from the outer heart surface and invade into the ventricle (V) chamber, as seen by the residual EGFP. (**j–l**) At 30 d.p.f., *careg:EGFP* is restricted to the outer layer of the ventricular wall, and is downregulated in the trabecular myocardium. *careg:EGFP* is also expressed in non-myocytes of the bulbus arteriosus (BA) and a few cells of the atrium (A). (**m–o**) Longitudinal sections of *careg:dmKO2;cmlc2:EGFP* transgenic heart at 30 d.p.f. immunostained for GFP (green) and dmKO2 (red). The *careg:dmKO2* transgenic line has a cardiac developmental expression pattern similar to that of *careg:EGFP* (Supplementary Fig. 10). N≥5.

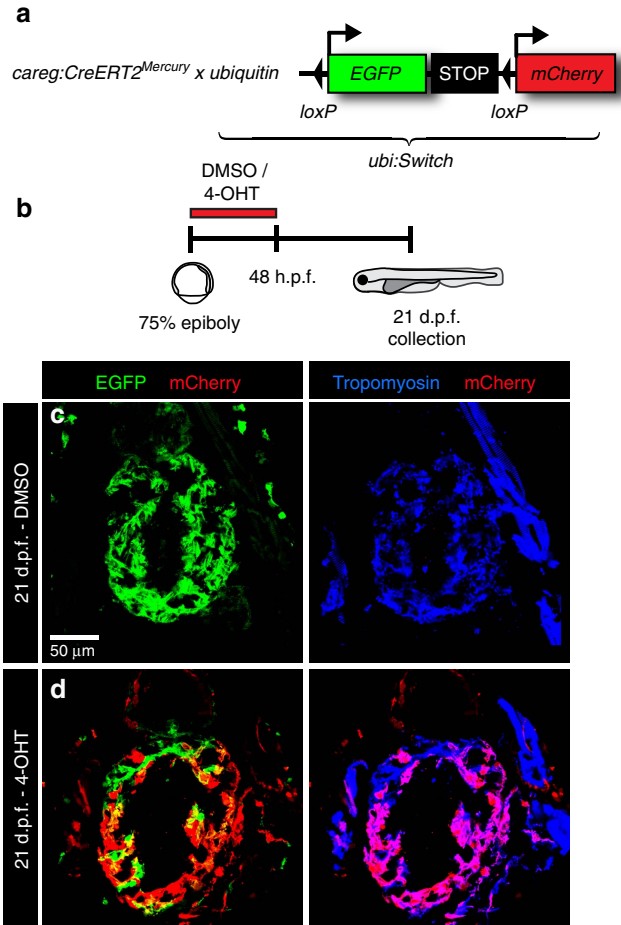

**Figure 7 | Embryonic *careg*-positive CMs contribute to the trabecular myocardium.** (**a**) Schematic representation of the transgenic strains used for lineage tracing. (**b**) Experimental design. (**c,d**) Longitudinal sections of hearts at 21 d.p.f. immunostained against mCherry (red), GFP (green) and Tropomyosin (blue). (**c**) Control embryos treated with the vehicle do not display mCherry fluorescence. (**d**) mCherry⁺ CMs are present in the trabecular and outer myocardium at 21 d.p.f. in embryos treated with 4-OHT, suggesting that the trabecular myocardium derives from embryonic *careg⁺* CMs. N = 9.

*careg*-expressing cells, especially in comparison to that of cortical CMs, which were typically rod-shaped with a very distinct striation (Fig. 8b). This suggests that *careg*-positive CMs possess less differentiated contractile structures. Second, *careg*-positive CMs displayed a slightly bent shape with their ends facing the ventricular chamber, suggesting the formation of attachments with trabecular CMs (Fig. 8b). To determine whether the *careg*-positive cell layer indeed interconnects with the trabecular myocardium, we performed antibody staining against N-cadherin. We found an enriched punctate pattern of N-cadherin immunostaining in *careg*-positive cells, facing the trabecular but not the cortical myocardium (Fig. 8d), suggesting an uneven amount of adhesion on both surfaces of these cells. Thus, *careg*-expressing CMs contain structural features of junctional/transitional CMs, forming the circumference of the trabecular myocardium.

Electron microscopy analysis has identified the presence of a fibroblast network in the junctional region[32]. To determine the spatial distribution of the primordial layer in this zone, we generated double transgenic fish *careg:dmKO2;wt1a-6.8kb:GFP*, the latter of which demarcates the subcortical non-myocytes[43,47]. We found that most *wt1a-6.8kb:GFP*-positive cells were arranged between the cortical myocardium and *careg*-labelled CMs (Fig. 8e). Thus, a fibroblast layer seems to 'glue' together the

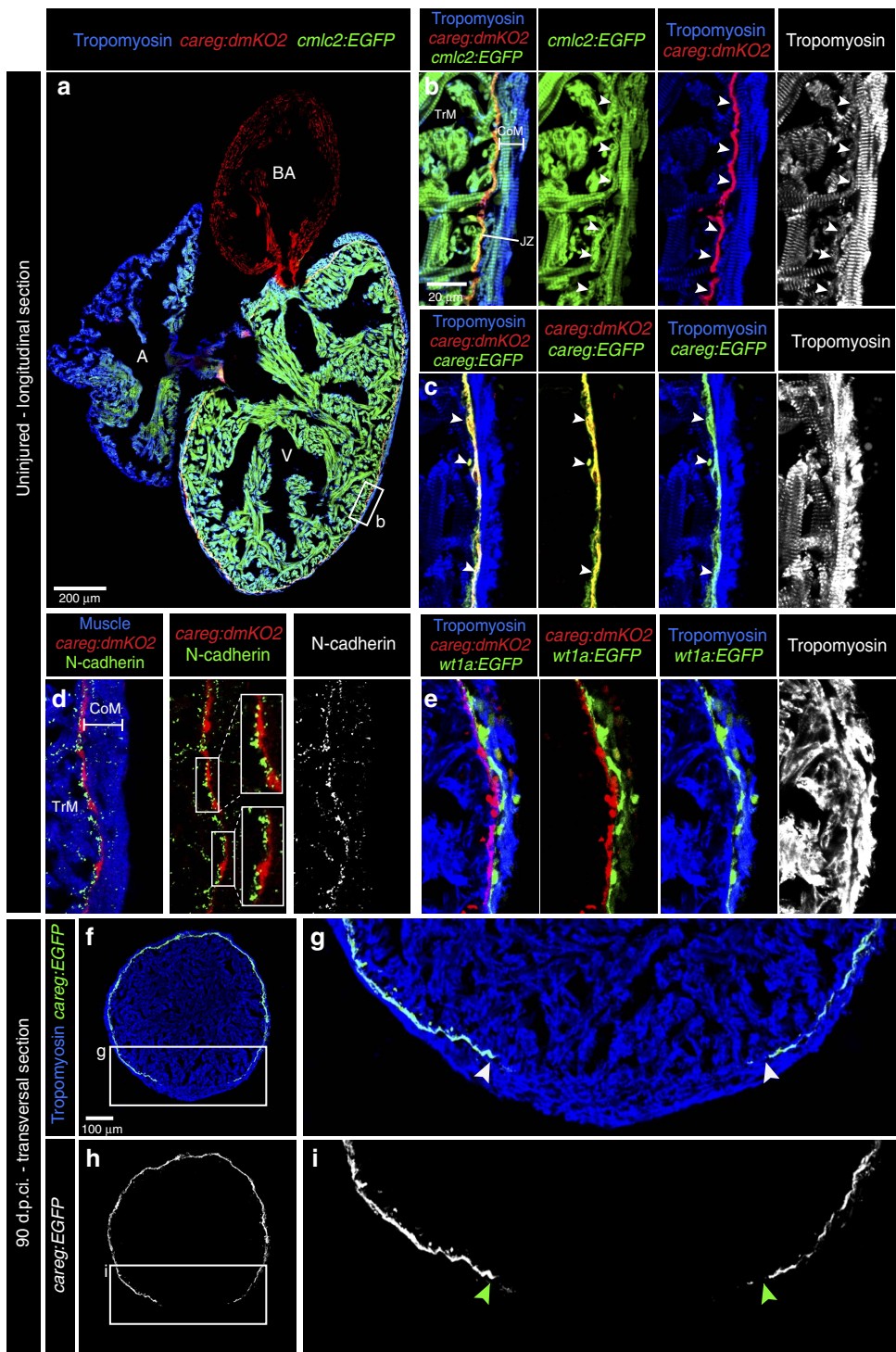

**Figure 8 | Monitoring the primordial CM layer between the compact and trabecular myocardium. (a)** Longitudinal sections of *careg:dmKO2;cmlc2:EGFP* transgenic adult hearts labelled with antibodies against KO2 (red) and Tropomyosin (blue, white). V, ventricle; A, atrium; BA, bulbus arteriosus. **(b)** *careg:dmKO2* expression is maintained during adulthood in a single layer of thin subcortical CMs (arrowheads) in the junctional zone (JZ) between the compact/cortical (CoM) and the trabecular myocardium (TrM). $N = 4$. **(c)** A higher magnification of the adult ventricle wall of double transgenic fish *careg:dmKO2; careg:EGFP* displays an overlapping expression of both markers in subcortical myocytes (arrowheads). $N = 4$. **(d)** A magnified adult ventricle wall of *careg:dmKO2* transgenic fish displays an abundant expression of N-cadherin (green, white) on the surface of the junctional CMs on the side facing the trabecular myocardium. $N = 4$. **(e)** A fragment of the adult ventricle wall of *careg:dmKO2;wt1a-6.8kb:GFP* transgenic fish displays the separation of compact myocardium and subcortical *careg:dmKO2*[+] layer by *wt1a-6.8kb:GFP*-labelled fibroblasts of the junctional zone. $N = 4$. **(f–i)** At 90 d.p.ci., transversal heart sections of *careg:EGFP* hearts display a gap in the primordial layer within the regenerated myocardium. The extent of the gap is indicated with arrowheads. $N = 4$.

cortical and primordial CMs. This suggests that the primordial layer is partially separated from the cortical layer by connective tissue.

**Incomplete regeneration of the primordial cardiac layer.** After injury, heart regeneration is completed within 30 to 60 days. Closer examination of *careg:EGFP* or *careg:dmKO2* transgenic fish revealed that the primordial/junctional CM layer was interrupted in the new myocardium, indicating its incomplete restoration, even at 90 d.p.ci. (Fig. 8f–i; Supplementary Fig. 3i,j). Thus, the regenerative competence of this CM layer is unexpectedly lower than that of other CMs.

To directly test the regenerative contribution of primordial/junctional *careg*-positive CMs, we performed lineage tracing of these cells. To this aim, we crossed *careg:CreERT2* with the *cmlc2:loxP-AmCyan-STOP-loxP-ZsYellow* lineage-tracing line, referred to as *cmlc2:Switch,* which labels cardiac cells in yellow after Cre-*lox* mediated recombination (Fig. 9a). In the absence of 4-OHT, no ZsYellow was detected in the ventricle sections of *careg:CreERT2;cmlc2:Switch* double transgenic fish (Supplementary Fig. 10a–c). To label the primordial CM layer, we pulsed adult transgenic fish *careg:CreERT2;cmlc2:Switch* with 4-OHT for 2 days, and we collected the hearts after 7 days (Supplementary Fig. 10d,e). We found that ∼30% (29.6 ± 2.5; $N = 6$) of the entire compact myocardium was demarcated by ZsYellow expression.

To determine whether primordial cells provide a source for the peri-injured trabecular myocardium, we performed cryoinjury 8 days after labelling of CMs, and analysed the hearts at 10 d.p.ci. (Fig. 9a,b). Although the primordial layer of the remaining myocardium displayed ZsYellow expression, no fluorescent labelling was detected in the peri-injury zone of the trabecular myocardium (Fig. 9c,d). These fate-mapping results indicate that *careg*-positive primordial CMs, which have been labelled in the intact heart, do not markedly contribute to the dedifferentiation zone in the regenerating heart.

To examine the contribution of the primordial/junctional CMs in the regenerated ventricles, we used the same experimental design (Fig. 9a,b), and analysed the hearts at 30 d.p.ci. We found the main portion of the new myocardium was not markedly labelled by zsYellow expression, suggesting a relatively low regenerative contribution of primordial/junctional CMs. Moreover, the outer wall of the new myocardium contained gaps of zsYellow-labelled cells (Fig. 9e,f). This lineage-tracing analysis indicates that the primordial/junctional cardiac layer is incompletely restored after regeneration, which is consistent with *careg:EGFP* or *careg:dmKO2* reporter results.

**Regenerative contribution of peri-injury CMs and MES.** To examine whether *careg* is induced *de novo* in the dedifferentiated trabecular CMs and to trace the destiny of *careg*-expressing cells during regeneration, we performed cell labelling after cryoinjury and monitored the distribution of the ZsYellow-positive CMs during new myocardium formation. To this aim, we performed cryoinjury of *careg:CreERT2;cmlc2:Switch* double transgenic fish, and gave animals a 4-OHT pulse between 5 and 7 d.p.ci. (Fig. 9g). Analysis of ventricle sections at 15 d.p.ci. revealed 55.1% ± 2.6 ($N = 5$) of ZsYellow-positive CMs in the peri-injury zone (Fig. 9h,i). At 30 d.p.ci., the injured tissue was replaced with a new myocardium that was abundantly labelled by ZsYellow-positive CMs within the trabecular and cortical regions (Fig. 9j,k). We concluded that the injury-abutting CMs from the trabecular and cortical myocardium were activated and provided a cellular source for the regenerated cardiac muscle. Thus, the lineage-tracing analysis revealed that non-primordial/junctional CMs

have a high regenerative capacity and contribute to new myocardium formation.

To examine whether *careg* is induced *de novo* in the activated stump mesenchyme of the fin, we performed cell labelling before and after amputation using *careg:CreERT2;ubi:Switch* transgenic fish. First, we gave animals a 1-day pulse of 4-OHT at 10 days before amputation, and monitored mCherry expression during regeneration (Supplementary Fig. 10f). Live-imaging and immunofluorescence analysis of fin sections did not reveal labelling, except for a few scattered cells (Supplementary Fig. 10g-p). However, when the animals were exposed to 4-OHT during the first day following amputation, the regenerative outgrowth was demarcated by mCherry, as assessed by live-imaging of the fins at different time points, and immunofluorescence analysis of longitudinal sections at 4 d.p.a. (Fig. 9l–u). We concluded that *careg* is induced in the pre-existing MES of the stump, which gives rise to the new MES of the regenerate. Taken together, the regeneration-competent functional cells of the zebrafish heart and fin activate the *careg* element, indicating a common regulatory mechanism of organ restoration that operates in distinct cell types, such as cardiac myocytes and fin fibroblasts.

**Discussion**

Organ regeneration involves a morphogenetic stimulation of the remaining tissue in the peri-injury zone. Our findings uncovered that the local regenerative response operates through common regulatory mechanisms in different body structures, such as cardiac and mesenchymal tissues. We identified a genetic element, called *careg*, which is transiently activated in the peri-injured CMs and fin MES during regeneration (Fig. 10). The identification of this transgenic biosensor supports the notion of regeneration-specific enhancers, which have been recently shown to act in the endocardium and the distal blastema[30]. Here we showed that during organ regeneration, the expression of *careg* in CMs and fin MES is dependent on TGFβ/Activin-β signalling. Previous studies from our laboratory have identified an enhanced expression of TGFβ/Activin-β ligands in wounded tissues of the heart and the fin, as well as the requirement of this pathway in organ restoration[36,42]. Remarkably, the *careg* reporter was not induced in all TGFβ/Activin-β-activated tissues after injury. In the heart, it was absent in the fibrotic area of the injured ventricle, despite the abundant number of pSmad3-positive cells and the requirement of TGFβ/Activin-β for the beneficial extracellular matrix deposition in this zone[42,43]. Thus, the responsiveness of the *careg* reporter to TGF-β signalling occurs specifically in regeneration-contributing cells and not in all regeneration-assisting cells of the injured heart. This finding indicates that *careg* should not be viewed as a universal responsive element of the TGFβ/Activin-β activity in zebrafish. We propose that the *careg* element represents a common biosensor that is triggered by TGFβ/Activin-β specifically in regenerating cells, such as CMs and MES, after injury of distinct zebrafish organs.

The *careg* element includes a 3.18 kb sequence immediately upstream of the *ctgfa* transcriptional start site. The bioinformatics analysis of this fragment indeed revealed a potential binding site for Smad3, but also for multiple other transcription factors. Thus, consistently with a previous study[37], the *careg* element may represent a combination of various enhancers, which require further biochemical and genetic examination. Although *careg* comprises the upstream sequence of the *ctgfa* gene, the latter is not upregulated in regenerating CMs and MES. This differential expression pattern suggests that the full endogenous genomic sequence might include potential regeneration-suppressor binding sites, which prevent *ctgfa* expression in activated

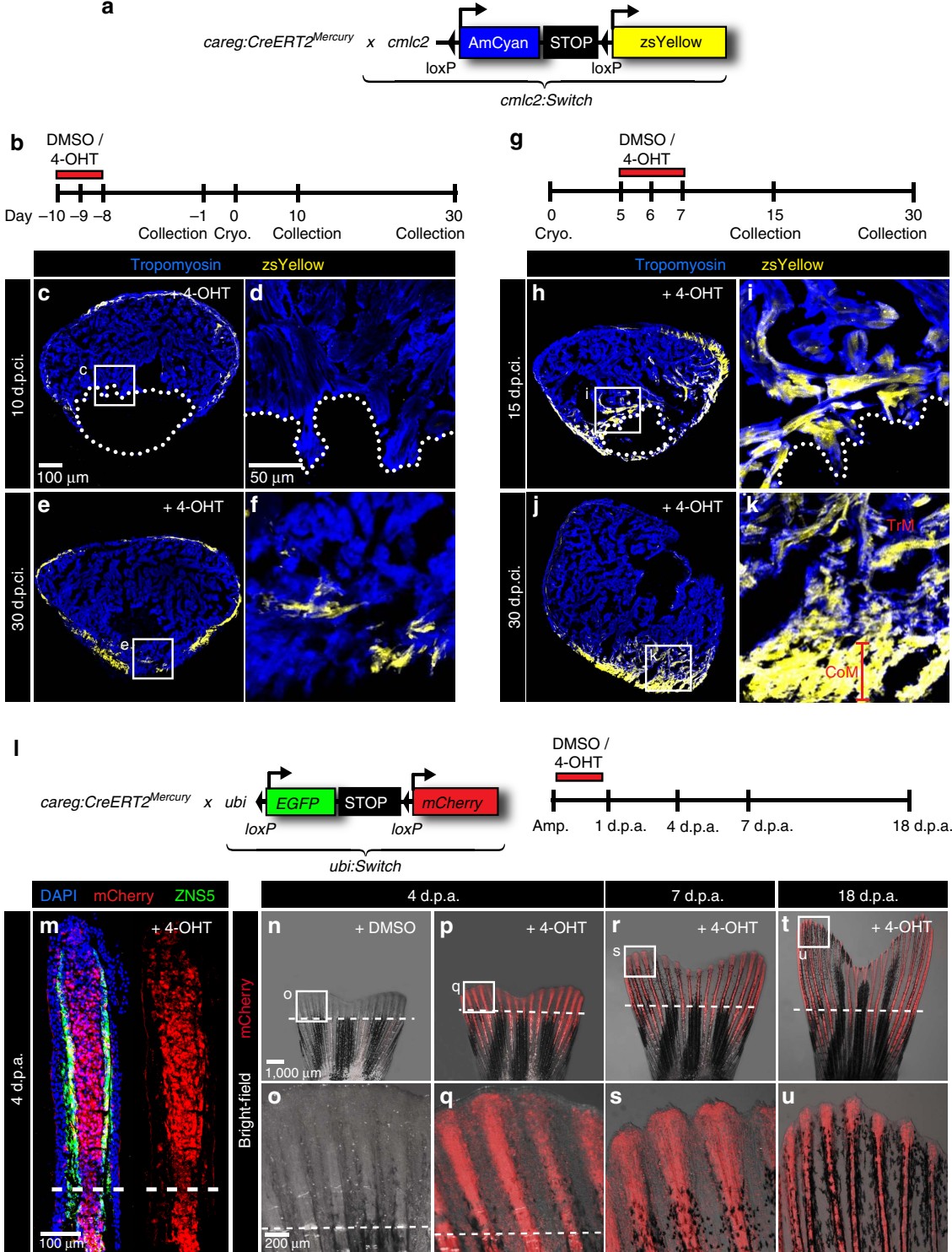

**Figure 9 | *careg*-expressing cells contribute to the regenerating myocardium and fin mesenchyme.** (**a,b**) Schematic representation of the transgenic strains and the experimental design for lineage tracing of *careg*-expressing primordial CMs during ventricle regeneration. (**c**–**f**) Representative sections of hearts treated with 4-OHT before cryoinjury to label the primordial layer, stained with antibody against Tropomyosin (blue). (**c,d**) At 10 d.p.ci., no zsYellow is detected in the trabecular myocardium at the injury site, indicating that primordial CMs do not contribute to the regeneration zone along the post-cryolesioned tissue. (**e,f**) At 30 d.p.ci., the zsYellow-positive primordial layer regenerates incompletely. (**g**) Experimental design for lineage tracing of *careg*-expressing dedifferentiated CMs during ventricle regeneration. (**h**–**k**) Representative sections of hearts treated with 4-OHT after cryoinjury hearts immunostained against Tropomyosin (blue) (**h,i**) At 15 d.p.ci., zsYellow is expressed in the trabecular myocardium at the injury site. (**j,k**). At 30 d.p.ci., zsYellow-positive CMs are detected in the trabecular (TrM) and cortical myocardium (CoM) of the regenerated muscle. $N \geq 8$. (**l**) Transgenic strains and the design of lineage tracing experiments during fin regeneration. (**m**) Longitudinal section of lineage-labelled fins at 3 d.p.a., stained for osteoblasts using the Zns5 antibody (green) and for nuclei with DAPI (blue). mCherry is detected in the entire mesenchyme of the regenerative outgrowth and below the amputation plane. $N = 6$. (**n**–**u**) Live-imaging of the same lineage-labelled fin at different time points of regeneration. The regenerative outgrowth and the stump tissue are labelled with mCherry. $N = 6$.

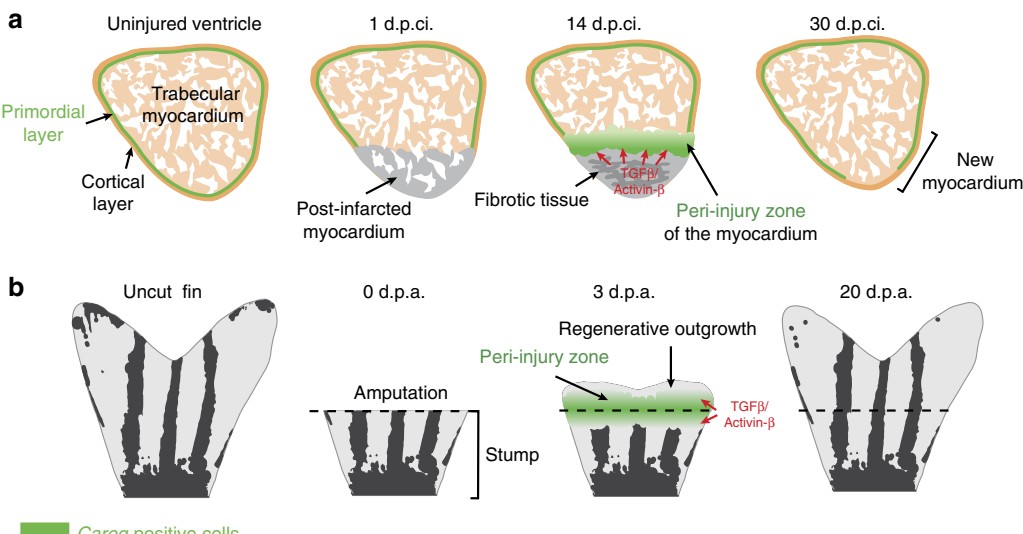

**Figure 10 | CMs and stump MES activate a common regulatory element during regeneration.** (**a,b**) A schematic representation of ventricle sections and whole fins at different time points during regeneration. The peri-injury zone of the ventricle and the stump of the fin activate a transient expression of the *careg* reporter in a TGFβ/Activin-β-dependent manner. In the heart, the *careg* reporter also demarcates the primordial layer, which fails to be completely restored in the new myocardium.

tissues. Further dissection of the adjacent genomic sequences beyond the *careg* element is required to identify potential negative regulatory elements. Indeed, a recent study on spinal cord regeneration established a *ctgfa:EGFP* transgenic fish line with a 5.5 kb sequence upstream of the *ctgfa* translational initiation codon[48]. It will be interesting to compare the expression of the *careg* and *ctgfa* reporters in the context of heart and fin regeneration.

Remarkably, the *careg* reporter is also constitutively expressed in the tissue-junctional layers, such as the primordial cardiac layer spanning the cortical and trabecular compartments, and the interface cells between bone matrix and MES in the fin. In the adult uninjured heart, our findings provide evidence that the primordial CM layer acts as a junctional/transitional layer, based on the architecture of the single-cell sheet, a flattened U-shaped cell morphology, an enrichment of N-cadherins facing the trabecular fascicles and the presence of overlying connective tissue. These interconnecting CMs are probably exposed to certain stimuli that trigger the activation of the same regulatory elements as during regeneration, although in a TGFβ/Activin-β independent manner. These stimuli remain unknown but they might be of biomechanical nature. Indeed, the junctional CM layer is thought to anchor the trabeculae with the luminal cavities in order to prevent compaction of the muscle[49,50]. In the locomotory appendage, osteoblasts connecting bones with MES may also cope with particular inter-tissue tension during swimming. Thus, the *careg* reporter is expressed in certain integrative cellular layers, which might be exposed to biophysical challenge during their normal function. Molecular mechanisms of such induction require further investigation.

The zebrafish ventricle comprises heterogeneous types of CMs[32,33]. Cortical CMs display a high regenerative potential, as previously shown using the *gata4*-reporter[13,21]. Our findings yielded several discoveries related to the regenerative competence of primordial and trabecular CMs. Lineage-tracing analysis in the embryo revealed that the primordial CM layer gives rise to the growing myocardium during development. In the adult heart, however, the primordial layer was not completely restored after cryoinjury, indicating its low ability to regenerate after traumatic injury. This limitation might be caused by the specialization of the primordial CM layer as the junctional layer of the heart wall.

By contrast, both the inner trabecular CMs, which build the main part of the ventricular chamber, and the cortical CMs, which surround the ventricular wall, all of them efficiently contribute to the formation of the new myocardium. It will be interesting to analyse *gata4/careg* double transgenic fish, to more accurately specify the relative contribution of cortical and trabecular CMs in the resection and cryoinjury models.

Our study of *careg* during fin regeneration allowed lineage tracing of the injury-activated MES. We show that MES of the regenerate arise from pre-existing local cells. This finding supports the concept that the newly formed tissue originates from the local stump and not from recruited remote cell populations[8,29]. The regenerative inadequacy in mammals is thought to result from the evolutionary loss of the intrinsic cellular plasticity that enables the activation of morphogenetic programs after injury. The identification of a common regulatory element in zebrafish heart and fin might provide a valuable resource for transgenic modulation of the regenerative response in other models.

## Methods
**Zebrafish lines and animal use.** Wild-type AB (Oregon) and transgenic female and male adult zebrafish aged 6 to 12 months were used in this study. Genetically modified lines were: *dob*[45], *Tg(ctgfa:EGFP)*[38], which we renamed *Tg(careg:EGFP)*, *Tg(cmlc2:DsRed2-Nuc)*[51], *Tg(cmlc2:EGFP)*[52], *Tg(cmlc2:loxP-AmCyan-loxP-ZsYellow)*, referred to as *cmlc2:Switch*[53], *ubi:loxP-EGFP-loxP-mCherry (ubi:Switch)*[54]; *Tg(wt1a-6.8kb:GFP)*[55].

To generate *careg* lines, 3.18 kb upstream regulatory sequence of *ctgfa*[37,38] was PCR-amplified from the plasmid pEGFP-1-ctgfa, which was kindly provided by Antonio Jacinto (Universidade Nova de Lisboa), with primers (F) 5′-CATGAGGAATGTTCCACTG-3′ and (R) 5′-GAGAAGCAACAGTCACTC GAC-3′. To establish *careg:dmKO2*[Kiwi], the PCR fragment was cloned into pDestTol2pA2 upstream of dmKO2 (ref. 39), kindly provided by Brian Link (Medical College of Wisconsin). To create *careg:CreERT2*[Mercury], the same PCR fragment of the *ctgfa* upstream regulatory sequence was cloned into p5E-MCS (#473, zcre.org). The pDest_crya:Venus_careg:CreERT2 construct was generated using multisite Gateway assembly of p5E-ctgfa, pMECreERT2 (#396, zcre.org), p3E_SV40polyA (#302, zcre.org) and pDestTol2crya:Venus (#480, zcre.org). Each plasmid was co-injected with the pCS2FA-transposase mRNA into one-cell-stage wild-type embryos.

All assays were performed using different animals that were randomly assigned to experimental groups. The exact sample size (N) was described for each experiment in the figure legends, and was chosen to ensure the reproducibility of the results. During invasive procedures and imaging, fish were anaesthetized with buffered solution of 0.6 mM tricaine (MS-222 ethyl-m-aminobenzoate, Sigma-

Aldrich) in system water. Immediately after the procedures, zebrafish were transferred into a tank with system water, and were continuously monitored until fish restarted swimming. Fish normally resumed breathing within 30 s after transfer into water. In cases when the breathing was not observed after this time, we stimulated the animals by squirting water into their gills with a plastic pipette until they started spontaneous breathing. During regeneration, fish were maintained at 26.5 °C. Animal procedures were approved by the cantonal veterinary office of Fribourg, Switzerland.

**Ventricular cryoinjury and resection.** Myocardial cryoinjuries were performed according to the video protocol[56]. Briefly, anaesthetized fish were placed ventral side up in a damp sponge under the stereomicroscope. To access the heart, a small incision (approximately 1–2 mm) was made through the chest with iridectomy scissors (Roboz Surgical Instrument Co.), a procedure described as thoracotomy[57]. The beating heart was well visible, and no extensive bleeding occurred during the thoracotomy. Then, the ventricular wall was directly frozen by applying for 23–25 s a stainless steel cryoprobe (custom-made) pre-cooled in liquid nitrogen. The tip of the cryoprobe was 6 mm long with a diameter of 0.8 mm, the handle of the cryoprobe was 4 cm long with a diameter of 8 mm and is covered with a plastic surface. To stop the freezing of the heart, system water at room temperature was dropped on the tip of the cryoprobe, and fish were immediately returned into a tank with water. It was not necessary to suture the incision of the chest, as the thoracic wound healed spontaneously within a week[57,58].

Ventricular apex resection was performed according to the established protocol[40]. Briefly, after thoracotomy, the apex of the ventricle was removed using curved iridectomy scissors.

To collect the heart for fixation, fish were killed in tricaine solution. A large (~4 mm) incision was made above the heart through the branchial cartilage and the heart was pulled from the body cavity as shown in the video protocol[56].

**Fin cryoinjury and amputation.** The fin cryoinjury was performed with a steel cryotome blade, (22 cm, 260 g, Product 14021660077, Leica). The cryotome blade was immerged for 90 s in liquid nitrogen, removed and held for 5 s in the air to avoid the dispersion of liquid nitrogen droplets. The pre-cooled blade was placed for 15 s on the fin surface perpendicularly to the proximo-distal axis of the appendage at an equidistant position between the fin base and the central cleft[41].

Fin amputations were performed using a razor blade. Adult caudal fins were amputated proximally to the first bone bifurcation point. Larval tails were cut at the posterior end of the notochord. Wound healing was spontaneous without significant bleeding. Live images of regenerating fins were taken with a Leica AF M205 FA stereomicroscope.

**Drug treatments.** The TGFβ type I receptor inhibitor SB431542 (Tocris) was dissolved in DMSO at a stock concentration of 20 mM and used at a final concentration of 20 µM. Control animals were kept in water with 0.1% DMSO. For lineage tracing, fish were incubated in 5 µM 4-hydroxytamoxifen (4-OHT, Tocris), which was made from a 10 mM stock solution dissolved in DMSO, for 24 or 48 h at the time points indicated. For BrdU incorporation, fish were incubated for 48 h in water containing 50 µg/ml BrdU (Sigma-Aldrich). Zebrafish were treated with drugs at a density of 3 fish per 100 ml of water, and then returned to circulating system water.

**Immunofluorescence analysis.** Embryos and fins were fixed in 4% paraformaldehyde (PFA), adult heart in 2% PFA, except the samples for N-cadherin immunostaining, which were fixed in Dent's fixative (80% methanol, 20% dimethylsulfoxide). The fixation was performed overnight at 4 °C, followed by washing in PBS (3 × 10 min each). Embryos and heart specimens were equilibrated in 30% sucrose at 4 °C. Fin specimens were first embedded in 1.5% agarose/5% sucrose blocks, and then equilibrated in 30% sucrose. Subsequently, all specimens were embedded in tissue freezing media (Tissue-Tek O.C.T.; Sakura) and cryosectioned at a thickness of 12–16 µm. Sections were collected on Superfrost Plus slides (Fisher) and allowed to air dry for ~1 h at room temperature (RT). The material was stored in tight boxes at −20 °C.

Before use, slides were brought to room temperature for 10 min, the area with sections was encircled with the ImmEdge Pen (Vector Laboratories) to keep liquid on the slides, and left for another 10 min at RT to dry. Then, the slides were transferred to coplin jars containing 0.3% Triton-X in PBS (PBST) for 10 min at RT. For BrdU and DsRed immunostainings, sections were incubated in 2 N HCl in PBST during 35 min and washed 3 × 10 min before the subsequent procedure.

The slides were transferred to a humid chamber. Blocking solution (5% goat serum in PBST) was applied on the sections for 1 h at RT. Subsequently, sections were covered with approx. 200 µl of primary antibody diluted in blocking solution and incubated overnight at 4 °C in the humid chamber. They were washed in PBST in coplin jars for 1 h at RT and again transferred to the humid chamber for incubation with secondary antibodies diluted in blocking solution. The slides were washed in PBST for 1 h at RT and mounted in 90% glycerol in 20 mM Tris pH 8 with 0.5% N-propyl gallate.

The following primary antibodies were used: chicken anti-GFP at 1:2,000 (GFP-1010, Aves Labs), rabbit anti-DsRed at 1:200 (632496, Clonetech), rabbit anti-KO2

1:200 (PM051M, Amalgaam) and mouse anti-KO2 1:500 (M168-3, Amalgaam), rabbit anti-mCherry 1:500 (GTX128508, GeneTex), mouse anti-embCMHC (N2.261) at 1:50 (developed by H.M. Blau, obtained from Developmental Studies Hybridoma Bank), mouse anti-tropomyosin at 1:100 (developed by J. Jung-Chin Lin and obtained from Developmental Studies Hybridoma Bank, CH1), rabbit anti-p-Smad3 at 1:400 (ab52903, Abcam), rat anti-BrdU at 1:100 (ab6326, Abcam), mouse anti-N-cadherin at 1:200 (610920, BD Biosciences), rabbit anti-Mef2c 1:50 (ab79436, Abcam), rabbit anti-Tenascin C 1:500 (T2550-23, US Biological); mouse Zns5 1:500 (ZDB-ATB-081002-37, ZFIN). The secondary antibodies (at 1:500) were Alexa conjugated (Jackson ImmunoResearch Laboratories). Phalloidin-Atto-565 (94072, Sigma) and Atto-647 N (65906, Sigma) were used at 1:500. DAPI (Sigma) was applied to label nuclei.

**In situ hybridization.** The following forward (F) and reverse (R) primers (5′ to 3′) were used to synthesize the template for the probes: ctgfa (NM_001015041) F: 5′-TGTGGCTCAAGAGTGCAGTG-3′ and R: 5′-CATGTCGCCAAC-CATCTTC-3′, egfp (pEGFP-1); F: 5′-ACGTAAACGGCCACAAGTTC-3′ and R: 5′-CTGGGTGCTCAGGTAGTGG-3′, msxB (NM_131260) F: 5′-GAGAAT-GGGACATGGTCAGG-3′ and R: 5′-GCGGTTCCTCAGGATAATAAC-3′. To synthesize the anti-sense probes, a sequence of T3 RNA polymerase promoter (5′-ATTAACCCTCACTAAAGGGAGA-3′) was added to the 5′ end of the reverse primers. The digoxigenin-RNA labelling mix (Roche) was used to generate the probe, which after purification was dissolved in hybridization solution and stored at −20 °C.

In situ hybridization was performed on heart and fin cryosections. Organs were fixed overnight at 4 °C in 4% PFA, then washed twice in PBS, once in PBS/methanol and in methanol for 10 min each, and finally transferred to methanol. Fixed material was stored at −20 °C until processed for cryosectioning, which was performed as for immunofluorescence. Before use, slides were brought to room temperature for 10 min, and incubated in hybridization solution (50% formamide, 5 × SSC, 1 × Denhardt's solution, 10% dextran sulphate, 0.1 mg ml⁻¹ yeast tRNA; all from Sigma-Aldrich) for 1 h at RT and for 1 h at 60 °C. Then, they were placed into tight boxes with wet paper towels, and 200 µl of probe diluted in hybridization solution (1:200) was applied on each slide that was then covered with a plastic coverslip (ApopTag, Merck Millipore) and placed in an incubator at 60 °C overnight. On the next day, slides were transferred to a coplin jar and soaked for 5 min with prewarmed 5 × SSC (saline-sodium citrate) to allow the removal of cover slips, and washed twice with 5 × SSC for 30 min and once with 0.2 × SSC for 1 h, each step at 60 °C. The final wash was done with 0.2 × SSC for 5 min at RT. Slides were allowed to dry for a few minutes at RT and sections was encircled with the ImmEdge Pen, and left for 15 min for drying. Slides were placed into humidified boxes, and 1 × blocking reagent solution (Roche) in maleic acid buffer (100 mM Maleic Acid, 150 mM NaCl, 0.2% Tween-20, pH 7.5; all from Sigma-Aldrich) was applied on sections for 1 h at RT. Subsequently, 250 µl of anti-dig-alkaline phosphatase (AP) antibody (Roche; diluted 1:4,000 in blocking reagent solution) was applied on sections for 2 h at RT. Slides were rinsed and washed twice for 30 min in maleic acid buffer in coplin jars. Then, slides were washed twice in AP-buffer (100 mM Tris HCL pH 9.5, 100 mM NaCl, 50 mM MgCl2, and 0.2% Tween-20) for 5 min. To prepare the staining reagents, NBT and BCIP compounds (Roche) were diluted in AP-buffer according to manufacturer's instruction with addition of 10% Polyvinyl alcohol (Sigma-Aldrich). The staining reaction was performed in a humidified box at 37 °C. The reaction was monitored under the microscope, and was stopped by washes in coplin jars with PBS for 15 min, 70% ethanol for 1 h, and 15 min in PBS. The slides with fins were mounted in Aquatex mounting medium (Merck Millipore).

For heart sections, after completion of in situ hybridization, we performed fluorescent immunostaining against cardiac Tropomyosin, in order to visualize the intact part of the myocardium in regenerating ventricles[59]. After immunofluorescence staining, slides were mounted in 90% glycerol in 20 mM Tris pH 8 with 0.5% N-propyl gallate. Immunofluorescence and corresponding bright-field images were taken using a Leica SP5 confocal microscope and a Zeiss microscope, respectively, and were superimposed using Adobe Photoshop.

**Image and statistical analysis.** Fluorescent images of sections were taken with a Leica confocal microscope (TCS SP5) and the image J 1.49c software was used for subsequent measurements. Live images of fins were taken with a Leica stereomicroscope. For quantification of positive/labelled nuclei, we calculated the proportion of immunostained cells per total number of DAPI-stained nuclei or cardiac nuclear marker using ImageJ 1.46r software. For quantification of positive CMs, we calculated the proportion of immunostained area per either Tropomyosin- or F-actin-labelled tissue. Error bars correspond to standard error of the mean (s.e.m.). Significance of differences was calculated using unpaired Student's t-test. Statistical analyses were performed with the GraphPad Prism software. All results are expressed as the mean ± s.e.m.

**Sequence and bioinformatical analysis.** An upstream genomic fragment of ctgfa was amplified from genomic DNA of wild-type AB zebrafish with the following primers: F: 5′-GAGCAGTGAGAGATGCATGG-3′ and R: 5′-CCTCCTGATCGT-

GTTGAGTG-3′. The PCR template was used for sequencing. The sequencing result was aligned to the *careg* sequence to verify the sequence similarity.

The LastZ sequence alignment programme[60] from ENSEMBL Compara[61] was used to build pairwise alignment of the *ctgfa* genomic region between zebrafish (*Danio rerio*, Ensembl version 87.10) and the following species: cavefish (*Astyanax mexicanus*), spotted gar (*Lepisosteus oculatus*), coelacanth (*Latimeria chalumnae*), tilapia (*Oreochromis niloticus*), platyfish (*Xiphophorus maculatus*), medaka (*Oryzias latipes*), fugu (*Takifugu rubripes*), stickleback (*Gasterosteus aculeatus*), tetraodon (*Tetraodon nigroviridis*), Xenopus (*Xenopus tropicalis*), chicken (*Gallus gallus*), mouse (*Mus musculus*), human (*Homo sapiens*).

Potential binding sites of transcription factors were predicted with the MatInsepctor software (MatInspector Release professional 8.3, December 2016, Genomatix Software Suite)[62].

**Data availability.** The authors declare that all data supporting the findings of this study are available within the article and its Supplementary Information files, or from the corresponding author upon reasonable request.

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

## Acknowledgements

We thank V. Zimmermann for excellent technical assistance and for fish care, A. Puoti, S. Käser-Pébernard and D. König for critical reading of the manuscript. We are grateful to Laurent Falquet, for help with the bioinformatics analysis; B. Link (Medical College of Wisconsin) for sending the dmKO2 plasmid; A. Jacinto (Universidade Nova de Lisboa) for sharing *ctgfa:EGFP* transgenic fish; C. Mosimann (University of Zürich) for providing *ubi:Switch* and *cmlc2:Switch* strains. This work was supported by the Swiss National Science Foundation, grant numbers: 310030_159995 and CRSII3_147675, and the Swiss Heart Foundation (Die Schweizerische Herzstiftung).

## Author contributions

C.P. planned, performed experiments and analysed the data. A.J. conceived and supervised the study, designed experiments and wrote the manuscript with input from C.P.

## Additional information

**Competing interests:** The authors declare no competing financial interests.

