## [Peer Review File · Nature Communications]

Reviewers' Comments:

Reviewer #1 (Remarks to the Author)

The paper by Catherine Pfefferli and Anna Jaźwińska describes the heart and fin injury expression pattern of an existing zebrafish reporter construct that had been analyzed previously in notochord development and some aspects of fin regeneration. The reporter gene, driven by a 3 kb sequence upstream of *ctgfa*, is expressed transiently in regenerating heart myocardial cells and blastemal cells of the regenerating fin. Interestingly, the reporter mRNA spatial expression pattern did not match the mRNA spatial expression pattern of *ctgfa*. Thus the authors call the regulatory element(s) within the 3 kb *ctgfa* promoter *careg* (*ctgf* reporter in regeneration). This non-association suggests that a regeneration-specific regulatory element(s) is encoded within *careg*, but these elements do not determine the spatial expression pattern of *ctgfa*, which the authors speculate is determined by additional regulatory elements. An independent transgenic line was created (with a different reporter) to show that the expression pattern is associated with *careg* and not the site of genomic integration. Expression of the *careg* reporter was also observed for other injury models although without assessment of *ctgfa* endogenous expression. Smad3 colocalization and use of an antagonist of Tgfb type I receptor signaling to show loss of *careg*-EGFP expression in cardiomyocytes and cells in the regenerating fin, support the idea that *careg* is a Tgfb/Smad3 regulatory element. The remainder of the study used lineage tracing to trace *careg*-expressing cells during heart development and regeneration, and during fin regeneration. These experiments provided several interesting descriptive insights about the developmental origin and regenerative competence of primordial and trabecular CMs.

The paper addresses a topic of interest to those working in the regenerative biology field. The experiments were done with adequate replication and appropriate statistical analyses were performed.

Major Concern:

This paper is a solid developmental biology contribution with nice imaging and lineage tracing, some new results mixed with results that confirm findings in previous cited work. However, the paper was set up to establish that a common regulatory element was used in two very different regeneration paradigms – fin MES and heart CMs. In this regard, we learned very little about the nature of this element – 3kb is a relatively large piece of DNA that was not interrogated genomically or epigenetically. The paper was a bit one-sided; far more results concerning *careg* in heart development and regeneration were presented relative to fin, giving an unbalanced feel to the paper. The take home message for fin was that local cells mediate regeneration, something that has been shown and is not controversial. Again, some nice developmental biology but not the pay-off expected from the title and introduction.

Minor Concerns:

I was a bit confused by *careg*-EGFP expression in *fgf20a* mutants. Earlier in the paper it was established that the *careg* reporter construct localized to blastemal cells...if *fgf20a* mutants fail to make a blastemal, within which cells is *careg* being expressed? This was not made clear.

Lines 202-203: "This finding suggests that the *careg* regulatory element represents a molecular marker of immature cardiac cells not only during regeneration but also during development." It is debatable that a presumptively de-differentiated cell during regeneration is "immature" in the same sense as an embryonic CM. Maybe focus wording in this paragraph on *careg*-EGFP/dmK02 expression in primordial CMs throughout development, and not attempt to associate with immaturity or insinuate de-differentiation.

It was suggested that *careg*-reporter expression in the fin was localized to fibroblasts but the

identity of the cell type was not established.

Reviewer #2 (Remarks to the Author)

Comments to authors:

It remains unknown whether regeneration of different organs utilizes common mechanisms. The authors identified a regulatory sequence of the *ctgta* gene that can drive reporter expression in both heart and fin regeneration. This regulatory sequence does not recapitulate endogenous *ctgf* expression but two independent reporter lines showed the same patterns, suggesting the expression is driven by the regulatory sequence instead of being influenced by the integration site in the genome. However, the authors were able to use this reporter line as a tool to mark the regenerative tissues and characterize primordial cardiomyocytes and mesenchymal cells in the fin. They further demonstrated that activation of this *ctgf* reporter is dependent on TGF β /Activin- β signaling. Using genetic lineage tracing with *ctgf* regulatory sequence to drive a CreER, the authors showed that the primordial/junctional cardiomyocytes do not contribute to heart regeneration. However, when lineage tracing was done after heart cryoinjury and fin amputation, the *ctgf* regulatory sequence marked cardiomyocytes and mesenchymal cells can contribute to both heart and fin regeneration. Overall the manuscript is well-written.

Major comments:

1. The authors made an important point that a common regenerative program is utilized for both fin and heart regeneration based on their observation that the *ctgf* regulatory sequence is activated during both fin and heart regeneration. However, this regulatory sequence does not reflect endogenous *ctgf* expression. Therefore, it is difficult to interpret the results. It is possible that this *ctgf* regulatory sequence contains a TGF β response element, and it is known that both fin and heart regeneration are regulated by TGF β signaling. Then the results presented here therefore are not that surprising.
2. The authors showed that the cells lineage traced by *ctgf* regulatory sequence contribute to both heart and fin regeneration. Are these cells required for regeneration? Using the *ctgf*:CreER line, the authors can try to ablate these cells to determine whether the fish fail to regenerate fins or hearts.

Minor comments:

1. It is not clear what the authors meant by "we searched an informative DNA regulatory element" (line80). It is also not clear how the authors focused on that transgenic reporter driven by the *ctgf* regulatory sequence. If this is a candidate approach, the authors can make it clear that they tested this reporter as a candidate since it was available.

Our response to Reviewers' comments:

We read reviews of our submission with great interest. We were gratified that both reviewers found substantial merit to our work that makes it potentially suitable for publication in Nature Communications. We have addressed the criticism of both reviewers, and we are submitting a revised manuscript for publication.

We thank both reviewers for their constructive comments, which improve our manuscript.

Reviewer #1 (Remarks to the Author):

*The paper by Catherine Pfefferli and Anna Jazwińska describes the heart and fin injury expression pattern of an existing zebrafish reporter construct that had been analyzed previously in notochord development and some aspects of fin regeneration. The reporter gene, driven by a 3 kb sequence upstream of *ctgfa*, is expressed transiently in regenerating heart myocardial cells and blastemal cells of the regenerating fin. Interestingly, the reporter mRNA spatial expression pattern did not match the mRNA spatial expression pattern of *ctgfa*. Thus the authors call the regulatory element(s) within the 3 kb *ctgfa* promoter *careg* (*ctgf* reporter in regeneration). This non-association suggests that a regeneration-specific regulatory element(s) is encoded within *careg*, but these elements do not determine the spatial expression pattern of *ctgfa*, which the authors speculate is determined by additional regulatory elements. An independent transgenic line was created (with a different reporter) to show that the expression pattern is associated with *careg* and not the site of genomic integration. Expression of the *careg* reporter was also observed for other injury models although without assessment of *ctgfa* endogenous expression. *Smad3* colocalization and use of an antagonist of *Tgfb* type I receptor signaling to show loss of *careg*-EGFP expression in cardiomyocytes and cells in the regenerating fin, support the idea that *careg* is a *Tgfb*/*Smad3* regulatory element. The remainder of the study used lineage tracing to trace *careg*-expressing cells during heart development and regeneration, and during fin regeneration. These experiments provided several interesting descriptive insights about the developmental origin and regenerative competence of primordial and trabecular CMs.*

The paper addresses a topic of interest to those working in the regenerative biology field. The experiments were done with adequate replication and appropriate statistical analyses were performed. Major concerns

Reviewer's concern:

This paper is a solid developmental biology contribution with nice imaging and lineage tracing, some new results mixed with results that confirm findings in previous cited work. However, the paper was set up to establish that a common regulatory element was used in two very different regeneration paradigms – fin MES and heart CMs. In this regard, we learned very little about the nature of this element – 3kb is a relatively large piece of DNA that was not interrogated genomically or epigenetically.

Our response: We agree with the reviewer that the nature of the *careg* element should be addressed. According to the reviewer's suggestion, we interrogated this element at the genomic level. We present our analysis in the new **Supplementary Fig. 5**. Firstly, we displayed the exact position of the genomic element that was used to generate the transgenic fish. Secondly, we performed LASTz net pairwise alignment of the *ctgfa* genomic region between zebrafish and 9 other fish species from distinct phylogenetic orders, as well as with *Xenopus*, chicken and 2 mammals (mouse and human). This analysis identified two conserved regions in the majority of the analysed animals. Thirdly, we applied

bioinformatics analysis to predict several transcription factor-binding sites within the entire *careg* element, including the indicated conserved regions. Importantly, a Smad3 binding site was identified in the *careg* element in addition to other sites.

We added the following sentences in the results and the discussion:

Page 7/8

“To determine the features of the *careg* sequence, we performed LASTz net pairwise alignment of the *ctgfa* genomic region between zebrafish and 9 other fish species from distinct phylogenetic orders, as well as with *Xenopus*, chicken, mouse and human (Supplementary Fig. 5a, b). This analysis identified two conserved regions within the *careg* sequence, a 150 bp fragment immediately before the transcription start site and a 400 bp region at approx. 1000 bp upstream from this position. The latter is conserved in 5 other fish species from different orders, namely cavefish, spotted, tilapia, platyfish and a lobe-finned fish *Coelacanth (Latimeria)*. Interestingly, this region was also present in *Xenopus* and chicken, but not in mammals or in certain fish, such as medaka, fugu, stickleback or tetraodon. Bioinformatics analysis predicted several different transcription factor-binding sites, such as Smad3, TCFE, MEF3, NKXH, GATA (Supplementary Fig. 5c and Supplementary Table 1). We concluded that the *careg* element consists of a combination of evolutionary conserved and unique sequences that might be regulated by multiple transcription factors.”

Page 16/17 (Discussion):

“The *careg* element includes a 3.18 kb sequence immediately upstream of the *ctgfa* transcriptional start site. The bioinformatics analysis of this fragment indeed revealed potential binding sites for Smads, but also for multiple other transcription factors. Thus, consistently with a previous study³⁶, the *careg* element may represent a combination of various enhancers, which require further biochemical and genetic examination. Although *careg* comprises the upstream sequence of the *ctgfa* gene, the latter is not upregulated in regenerating CMs and MES. This differential expression pattern suggests that the full endogenous genomic sequence might include potential regeneration-suppressors, which prevent *ctgfa* expression in activated tissues. Further dissection of the adjacent genomic sequences beyond the *careg* element is required to identify potential negative regulatory elements. Indeed, a recent collaborative study on spinal cord regeneration from K. Poss and D. Stainier laboratories established a *ctgfa:EGFP* transgenic fish line with a 5.5-kb sequence upstream of the *ctgfa* translational initiation codon⁴⁶. It will be interesting to compare the expression of the *careg* and *ctgfa* reporters in the context of heart and fin regeneration.”

Reviewer's concern:

*The paper was a bit one-sided; far more results concerning *careg* in heart development and regeneration were presented relative to fin, giving an unbalanced feel to the paper. The take home message for fin was that local cells mediate regeneration, something that has been shown and is not controversial. Again, some nice developmental biology but not the pay-off expected from the title and introduction.*

Our response: We agree with the reviewer, that the manuscript more extensively addresses the heart than the fin. This imbalance was due to the detailed characterization of the primordial cardiac layer at the developmental, homeostatic and regenerative levels. We included this part because the *careg* reporter provides, to our knowledge, the first marker of this peculiar cardiac structure. Thus, we believe that this part of our study brings a valuable contribution to the cardiac regeneration and developmental field.

To increase the significance of the paper for the fin part, we now added three new experiments.

1.

In the previous version, the injury models used for both organs were not balanced. For the heart, we had applied the cryoinjury procedure as the main injury model, but in a supplementary figure, we had also demonstrated that the *careg* reporter was activated after ventricular apex amputation. For the fin, we had showed only one amputation model.

In the revised manuscript, we added an experiment with the cryoinjury procedure of the fin to reveal that the *careg* reporter is activated irrespectively of the type of injury not only in the heart but also in the fin. We show this finding in a new main figure **Fig. 3b-f**.

To further strengthen this experiment, we performed in-situ hybridization of cryoinjured fins using *ctgfa* probe, and *msxb* as a positive control of the activated mesenchyme. Cryoinjured fins at two time points (5 and 7 dpci) did not display *ctgfa* expression. Thus, as in amputation model, the expression of the reporter was not associated with the induction of the endogenous *ctgfa* gene in the cryoinjury model (new **Supplementary Fig. 6a-d**).

We added the following text in the results on the page 8:

“As opposed to the fin amputation model, the non-surgical exposure to the cold blade along the cryoinjury plane results in a progressive and spontaneous tissue detachment that is apparent between 12 and 48 hours after procedure³⁹. Analysis of fins shortly after cryoinjury did not reveal any expression of *careg:EGFP* (Fig. 3b). During shedding of the damaged distal part, at 1 and 2 dpci, *careg:EGFP* was weakly induced in the stump, suggesting that the clearance of dead tissues was associated with a concomitant activation of the regenerative program in the remaining cells (Fig. 3c-d). Remarkably, the expression of *careg:EGFP* became further upregulated at 4 dpci, when the regenerative process was resumed after reparation of the distorted margin (Fig. 3e). As in amputated fins, *in-situ* hybridization on longitudinal fin sections showed that the expression of the reporter was not associated with the induction of the endogenous *ctgfa* gene (Supplementary Fig. 6a-d). At the advanced regenerative phase, at 14 dpci, *careg:EGFP* was downregulated, consistent with the amputation model (Fig. 3f).”

2.

Second, we addressed the question about the role of TGFβ/Activin-β signalling for the homeostatic expression of the *careg* reporter in both organs.

In the first version, for the heart, we had shown that *careg* expression in the primordial cardiac layer was not affected by the prolonged treatment with the inhibitor of TGFβ/Activin-β signalling (former Fig. 4), but we had not performed a complementary experiment for the fin.

In the revised manuscript, we analysed the *careg* expression in uninjured fins after 7 days of inhibitor treatment. We found that *careg* expression in the osteoblasts adjacent to mesenchyme remained unaffected in this experiment. We included a new supplementary figure to show side by side that the homeostatic expression of *careg* is not dependent on TGFβ/Activin-β signalling both in the heart and the fin (new **Supplementary Fig. 7**).

3.

Third, we explored the *careg* reporter to learn more about the molecular and cellular mechanisms related to regeneration of both organs.

In the first version, for the heart, we had described the cellular contribution of new myocardium. Specifically, we had shown for the first time that the trabecular myocardium, in addition to the compact myocardium, gave rise to new cardiac muscle cells, while the primordial cardiac layer unexpectedly had a low regenerative potential.

In the revised manuscript, we would like to balance this discovery by focusing on the molecular network that promotes blastema formation. The examination of the *careg* reporter in combination with the pSmad3 immunofluorescence in the *fgf20a* mutant background revealed the TGF β /Activin- β activity is independent of Fgf20a signalling during fin regeneration.

We added the following paragraph to the text:

Page 10:

“Although Fgf20a and TGF β /Activin- β signalling have been described as the early inducers of blastema formation, the interaction between both pathways remains unknown. Using the *careg* element and pSmad3 immunofluorescence, we addressed a question whether the function of the TGF β /Activin- β signalling is dependent on Fgf20a. Our analysis of longitudinal fin sections revealed that the pSmad3 activity was blocked after SB431542 treatment, but remained unchanged in *dob* mutant fins (Fig. 4h-j). This finding indicates that TGF β /Activin- β acts independently of the Fgf20a activity during fin regeneration, providing a new insight into the molecular network guiding blastema formation.”

To further support this experiment, we added a new experiment with an early time point, at 1 dpa, shown in a new **Supplementary Fig. 9**.

Minor Concerns:

Reviewer's concern:

*I was a bit confused by *careg*-EGFP expression in *fgf20a* mutants. Earlier in the paper it was established that the *careg* reporter construct localized to blastemal cells....if *fgf20a* mutants fail to make a blastemal, within which cells is *careg* being expressed? This was not made clear.*

Our response: Indeed, the expression of *careg:EGFP* earlier in the paper should be described more precisely. We added the following explanation in the first section of the results:

Page 6:

“At 1 and 3 dpa, the regeneration zone included not only the outgrowing blastema, but also the peri-injury stump tissue within a distance of three ray segments below the amputation plane. Immunofluorescence analysis of longitudinal fin sections at 3 dpa showed that the expression of *ctgfa:EGFP* was induced in the Tenascin-C-positive MES of the stump and the blastema (Fig. 1h).”

Reviewer's concern:

*Lines 202-203: “This finding suggests that the *careg* regulatory element represents a molecular marker of immature cardiac cells not only during regeneration but also during development.” It is debatable that a presumptively de-differentiated cell during regeneration is “immature” in the same sense as an embryonic CM. Maybe focus wording in this paragraph on *careg*-EGFP/dmK02 expression in primordial CMs throughout development, and not attempt to associate with immaturity or insinuate de-differentiation.*

Our response: To avoid association between regenerating and embryonic CMs, we reformulated this

sentence:

“This finding suggests that the *careg* regulatory element represents a molecular marker of **activated cardiac cells during regeneration and immature CMs during development.**”

Reviewer's concern:

*It was suggested that *careg*-reporter expression in the fin was localized to fibroblasts but the identity of the cell type was not established.*

Our response: We agree that unambiguous markers for less specialized cells, such as mesenchymal cells/fibroblasts, need to be established. This will require a search for specific epitopes/antibodies, which is an important future goal. In the meantime, we can rely on many published studies that precisely characterize the distribution of tissues in the fin.

We observed that during regeneration, the *careg* reporter was induced in the spindle-like cells filling the space between the bones of the rays (intraray region) (Fig. 2g, g'). According to the current knowledge, the intraray tissue is composed of vascularized and innervated connective tissue that is built by fibroblasts/mesenchymal cells (reviewed in Tormini and Poss 2014 Dev. Cell; Jazwinska and Sallin 2016, J of Pathology). We used osteoblast *Zns5* marker to identify bone-producing cells (Fig. 1j). As the fin lacks other specialized cells, such as chondrocytes, myofibroblasts or muscle cells, the mesenchymal identity of the fin intraray tissue is considered as evident. In our study, we applied Tenascin C, as an extracellular matrix protein deposited by fin fibroblasts during regeneration (Fig. 1h and Fig. 2g, g') (Jazwinska et al., 2007 Current Biol). Further cell identity markers should be established in the future to more distinctively label zebrafish fibroblasts.

--

Reviewer #2 (Remarks to the Author):

Comments to authors:

*It remains unknown whether regeneration of different organs utilizes common mechanisms. The authors identified a regulatory sequence of the *ctgta* gene that can drive reporter expression in both heart and fin regeneration. This regulatory sequence does not recapitulate endogenous *ctgf* expression but two independent reporter lines showed the same patterns, suggesting the expression is driven by the regulatory sequence instead of being influenced by the integration site in the genome. However, the authors were able to use this reporter line as a tool to mark the regenerative tissues and characterize primordial cardiomyocytes and mesenchymal cells in the fin. They further demonstrated that activation of this *ctgf* reporter is dependent on TGFb/Activin-b signaling. Using genetic lineage tracing with *ctgf* regulatory sequence to drive a CreER, the authors showed that the primordial/junctional cardiomyocytes do not contribute to heart regeneration. However, when lineage tracing was done after heart cryoinjury and fin amputation, the *ctgf* regulatory sequence marked cardiomyocytes and mesenchymal cells can contribute to both heart and fin regeneration. Overall the manuscript is well-written.*

Major comments:

Reviewer's concern:

*1. The authors made an important point that a common regenerative program is utilized for both fin and heart regeneration based on their observation that the *ctgf* regulatory sequence is activated during both fin and heart regeneration. However, this regulatory sequence does not reflect endogenous *ctgf* expression. Therefore, it is difficult to interpret the results.*

It is possible that this ctgf regulatory sequence contains a TGF beta response element, and it is known that both fin and heart regeneration are regulated by TGF beta signaling. Then the results presented here therefore are not that surprising.

Our response:

We agree with the reviewer that this regulatory sequence most likely contains TGF-beta response element. We have now added a new **Supplementary Fig. 5** and **Supplementary Table 1**, which indicate a predicted Smad3-binding site using MatInspector software (Genomatix). Interestingly, the comparison between TGF-beta activity and *careg* expression suggest that the nature of the reporter is more complex than a universal TGF beta response element. Specifically, the fibrotic tissue, which transiently repairs the damaged myocardium, does not display *careg* expression despite of the abundant TGF-beta activity in this tissue. Thus, the responsiveness of the *careg* reporter to TGF-beta signalling occurs specifically in regeneration-participating cells and not in regeneration-assisting cells. This unexpected finding requires further biochemical evidence to fully understand the regulation of the *careg* element during regeneration and homeostasis.

To better clarify this point we added the following explanation in the discussion:

Page 16:

“Remarkably, the *careg* reporter was not induced in all TGFβ/Activin-β-activated tissues after injury. In the heart, it was absent in the fibrotic area of the injured ventricle, despite the abundant number of pSmad3-positive cells and the requirement of TGFβ/Activin-β for the beneficial extracellular matrix deposition in this zone^{40,45}. Thus, the responsiveness of the *careg* reporter to TGFβ/Activin-β signalling occurs specifically in regeneration-contributing cells and not in all regeneration-assisting cells of the injured heart. This finding indicates that *careg* should not be viewed as a universal responsive element of the TGFβ/Activin-β activity in zebrafish. We propose that the *careg* element represents a common biosensor that is triggered by TGFβ/Activin-β specifically in regenerating cells, such as CMs and MES, after injury of distinct zebrafish organs.

Next, we addressed the reviewer’s comment that the *careg* reporter did not reflect endogenous *ctgfa* expression, which might be confusing. To clarify this issue, we added the following explanation in the Discussion.

Page 16:

“The *careg* element includes a 3.18 kb sequence immediately upstream of the *ctgfa* transcriptional start site. The bioinformatics analysis of this fragment indeed revealed a potential binding site for Smads, but also for multiple other transcription factors. Thus, consistently with a previous study³⁶, the *careg* element may represent a combination of various enhancers, which require further biochemical and genetic examination. Although *careg* comprises the upstream sequence of the *ctgfa* gene, the latter is not upregulated in regenerating CMs and MES. This differential expression pattern suggests that the full endogenous genomic sequence might include unknown regeneration-suppressors, which prevent *ctgfa* expression in activated tissues. Further dissection of the adjacent genomic sequences beyond the *careg* element is required to identify potential negative regulatory elements. Indeed, a recent collaborative study on spinal cord regeneration from K. Poss and D. Stainier laboratories established a *ctgfa:EGFP* transgenic fish line with a 5.5-kb sequence upstream of the *ctgfa* translational initiation codon⁴⁶. It will be interesting to compare the expression of the *careg* and *ctgfa* reporters in the context of heart and fin regeneration.”

Reviewer’s concern:

2. The authors showed that the cells lineage traced by *ctgf* regulatory sequence contribute to both heart

and fin regeneration. Are these cells required for regeneration? Using the ctgf:CreER line, the authors can try to ablate these cells to determine whether the fish fail to regenerate fins or hearts.

Our response:

Yes, we have generated a new transgenic fish strains to address this question: *ctgf:Nitroreductase-mCherry*. Unfortunately, the ablation of *ctgf*-positive cells resulted in the lethality of fish within 1-3 days. We concluded that *ctgf*-expressing cells are indispensable for survival of the fish, and thus, most likely for regeneration. We did not include these results in the manuscript.

Minor comments:

Reviewer's concern:

1. It is not clear what the authors meant by “we searched an informative DNA regulatory element” (line80). It is also not clear how the authors focused on that transgenic reporter driven by the ctgf regulatory sequence. If this is a candidate approach, the authors can make it clear that they tested this reporter as a candidate since it was available.

Our response:

We modified this sentence according to the reviewer suggestion.

Our previous statement: “we searched for an informative DNA regulatory element” was replaced with “we used a candidate approach to identify an informative DNA regulatory element”.

Reviewers' Comments:

Reviewer #1 (Remarks to the Author)

I appreciate that the authors addressed all reviewer concerns. This is a very nice contribution that will influence thinking in the field of regenerative biology.

Reviewer #2 (Remarks to the Author)

The authors have addressed most of my previous comments except #2:

"The authors showed that the cells lineage traced by *ctgf* regulatory sequence contribute to both heart and fin regeneration. Are these cells required for regeneration? Using the *ctgf:CreER* line, the authors can try to ablate these cells to determine whether the fish fail to regenerate fins or hearts."

The authors stated that they used the *careg:nitroreductase-Cherry* to address this question. Unfortunately, *careg:nitroreductase-Cherry* fish die within 3 days. I assume that fish died after ablating *careg*-expressing cells during heart regeneration. Otherwise, this will be a critical experiment. However, it is unlikely ablating *careg* expressing cells after fin amputation will cause lethality in fish. Can author address the requirement of *cqreg* expressing cells in fin regeneration?

Response to the last comment of the reviewer

Reviewer comment:

The authors have addressed most of my previous comments except #2:

"The authors showed that the cells lineage traced by ctgf regulatory sequence contribute to both heart and fin regeneration. Are these cells required for regeneration? Using the ctgf:CreER line, the authors can try to ablate these cells to determine whether the fish fail to regenerate fins or hearts."

The authors stated that they used the careg:nitroreductase-Cherry to address this question. Unfortunately, careg:nitroreductase-Cherry fish die within 3 days. I assume that fish died after ablating careg-expressing cells during heart regeneration. Otherwise, this will be a critical experiment. However, it is unlikely ablating careg expressing cells after fin amputation will cause lethality in fish. Can author address the requirement of careg expressing cells in fin regeneration?

Our answer:

Although the *careg* reporter is induced in regenerating cardiomyocytes and fin mesenchyme, it is also expressed in certain adult tissues during homeostasis, such as cells of the vertebral bodies/discs and a subset of arteries. As the integrity of these structures is life-essential, we think that their genetic ablation might lead to fish lethality. Due to this limitation, the experiments proposed by the reviewer are currently not feasible. To address the requirement of *careg*-positive cells in the regeneration context, we are currently aiming to develop a new genetic tool for functional analysis of the *careg* reporter specifically during regeneration.